# Automatic Delineation of Cracks with Sentinel-1 Interferometry for Monitoring Ice Shelf Damage and Calving

Ludivine Libert[1], Jan Wuite[1], Thomas Nagler[1]

[1]ENVEO IT GmbH, Innsbruck, 6020, Austria

*Correspondence to*: Ludivine Libert (ludivine.libert@enveo.at)

**Abstract.** Monitoring the evolution of ice shelf damage such as crevasses and rifts is important for a better understanding of the mechanisms controlling the breakup of ice shelves and for improving predictions about iceberg calving and ice shelf disintegration. Nowadays, the previously existing observational gap has been reduced by the Copernicus Sentinel-1 Synthetic Aperture Radar (SAR) mission that provides a continuous coverage of the Antarctic margins with a 6 or 12-day

repeat period. The unprecedented coverage and temporal sampling enables for the first time a year-round systematic monitoring of ice shelf fracturing and iceberg calving, as well as the detection of precursor signs of calving events. In this paper, a novel method based on SAR interferometry is presented for an automatic detection and delineation of active cracks on ice shelves. Propagating cracks cause phase discontinuities that are extracted automatically by applying a Canny edge detection procedure to the spatial phase gradient derived from a SAR interferogram. The potential of the proposed method is

demonstrated in the case of Brunt Ice Shelf, Antarctica, using a stack of 6-day repeat-pass Sentinel-1 interferograms acquired between September 2020 and March 2021. The full life cycle of the North Rift is monitored, including the rift detection, its propagation at rates varying between 0.25 km d$^{-1}$ and 1.30 km d$^{-1}$, and the final calving event that gave birth to the iceberg A74 on 26 February 2021. The automatically delineated cracks agree well with the North Rift location in Landsat-8 images and with the eventual location of the ice shelf edge after the iceberg broke off. The strain variations

observed in the interferograms are attributed to a rigid-body rotation of the ice about the expanding tip of the North Rift in response to the rifting activity. The extent of the North Rift is captured by SAR interferometry well before it becomes visible in SAR backscatter images and a few days before it could be identified in optical images, hence highlighting the high sensitivity of SAR interferometry to small variations in the ice shelf strain pattern and its potential for detecting early signs of natural calving events, ice shelf fracturing and damage development.

**1 Introduction**

Because of their buttressing effect that regulates the upstream flow of the grounded ice sheet, ice shelves play a key role in the mass balance of the Antarctic ice sheet. Ice shelf calving, especially for ice shelves that originate from large tributary glaciers, constitutes one of the main contributions to the mass loss in Antarctica (the IMBIE team, 2018; Rignot et al., 2019). However, predictions about this contribution remain uncertain because of the poor understanding of the mechanisms

controlling ice shelf break up and the difficulty to model them (Sun et al., 2017; Cook et al., 2018), partly due to a lack of comprehensive observational data. Ice shelves can be structurally weakened by processes such as ice shelf thinning, that leads to ice flow speedup and increased shearing, or such as ice shelf retreat that results in unpinning. These processes may trigger a feedback response, thereby enhancing damage such as deeply crevassed areas and open fractures, increasing the ice velocity gradient and further weakening the ice shelf structure (Lhermitte et al., 2020). The complex response of an ice shelf

to rifting, the difficulty to predict ice shelf disintegration and the resulting uncertainties in mass balance models highlight the need for systematic monitoring of the damage evolution (Pattyn et al., 2017).

On-site measurements of ice shelves and active rifts from e.g. ground penetrating radar, time-lapse camera or GPS provide valuable insights for monitoring damage and better understanding the mechanisms leading to rift propagation (Banwell et al., 2017; King et al., 2018; De Rydt et al., 2019). However, field missions are expensive, necessitate heavy logistics and only

focus on a specific area (usually close to a base station) for limited periods in time. Therefore, despite their unquestionable value, they provide no feasible solution for continuous long-term and large-scale monitoring of ice shelf rifting systems.

Nowadays, most of the Antarctic ice shelves are routinely monitored with optical and radar satellites, providing dense image time series that enable the continuous observation of fracture opening, propagation, widening and iceberg calving in near real-time. For damage monitoring, Synthetic Aperture Radar (SAR) sensors constitute a good alternative to optical satellite

imaging thanks to their all-day/all-weather/year-round observing capability. Compared to optical images, SAR backscatter imagery presents also the advantage of the signal penetration through dry snow, making sub-surface crevasses and snow-filled fractures visible. In particular, studies like e.g. Thompson et al. (2020) or Marsh et al. (2021) report on the potential of TerraSAR-X high resolution Stripmap and Spotlight imagery for the identification of ice shelf cracks and crevasses as narrow as a few centimeters width. They also underline the strong dependence of the crevasses visibility on the feature

orientation, the acquisition geometry (look direction and incidence angle) and the snowpack water content that may prevent signal penetration and observation of deeply buried features. Unfortunately, TerraSAR-X high resolution images only cover small regions (typically $10 \times 10$ km for Spotlight mode, $30 \times 50$ km for Stripmap) and are not systematically acquired over Antarctica. In contrast, the acquisition strategy of Sentinel-1 provides a continuous coverage of almost the entire ice sheet margin of Antarctica with 6- and 12-day repeat intervals, which enables the systematic surveillance of ice shelf fracturing

with radar imaging for the first time (Torres et al., 2016).

Previous surveillance of cracks with satellite imagery was performed through visual inspection and only few studies investigated automatic methods for mapping the fracturing of ice shelves. Moctezuma-Flores and Parmiggiani (2016) proposed the use of a morphological filter for reducing speckle noise in SAR backscatter data, followed by a stochastic segmentation for mapping the pre-collapse fractured area of Nansen Ice Shelf. However, this approach was applied on a

subset of the SAR image focusing on a widely opened fracture. In practice, edge detection performed on wide swath SAR images often misses thin cracks and provides no distinction between topographic features like calving fronts, crevasses or rifts, if no contextual information is used.

Aside SAR backscatter imaging, a few studies reported on the potential of SAR interferometry (InSAR) for mapping rifting activity (Rignot and MacAyeal, 1998; Larour et al., 2004; Hogg and Gudmundsson, 2017; De Rydt et al., 2018). These studies showed that, in an interferogram, opened fractures correspond to well-defined and visually identifiable phase discontinuities. Rignot and MacAyeal (1998) identified rifts as branch-cut discontinuities and interpreted the fringe patterns over downstream ice shelf fragments as due to a rigid-body rotation about an axis perpendicular to the ice shelf surface and located at the tip of the rift. The analysis of double difference interferograms and the modelling efforts presented in the complementary paper by MacAyeal et al. (1998) support the hypothesis that this rigid-body rotation originates from creep flow. At the time of writing, to our knowledge, no study proposed a method for extracting the crack location automatically from the complex phase information supplied by an interferogram.

In this paper, we present an automatic method for delineating ice shelf fractures using Sentinel-1 Interferometric Wide SAR interferometry (Yague-Martinez et al., 2016). The proposed method exploits the ice deformations caused by the changing stress field and the shearing of the ice flow caused by the rifting activity, which translate into a discontinuous fringe pattern in an interferogram. We show that an active crack separates an ice shelf into distinct regions, characterized by fringe patterns with different orientations and different fringe rates that can be quantitatively derived by calculating the phase gradient, and that active cracks correspond to spatial phase discontinuities that can be mapped with an edge detection procedure. Because Brunt Ice Shelf (BIS) showed significant rifting activity in the past years (De Rydt et al., 2019), we select it as test site. The performance of the method is demonstrated using a set of 6-day repeat-pass Sentinel-1 interferograms acquired over BIS between September 2020 and March 2021. In particular, we track the activation and the propagation of a new rift, the North Rift, that led to the calving of iceberg A74 on 26 February 2021. Based on this study case, we demonstrate that SAR interferometry is sensitive to the dynamical response of an ice shelf to rifting activity and has potential to provide early indications of fracturing, not yet visible in SAR backscatter or optical satellite images.

In Section 2, the BIS test site is briefly described and, in Section 3, the ability of InSAR at capturing rifting activity is introduced. The delineation method and the processing line are described in Section 4, along with illustrating examples of intermediate processing steps. The capabilities and limitations of the method are also discussed. The Sentinel-1 dataset used for tracking the North Rift propagation on BIS and the processing parameters are detailed in Section 5. In Section 6, we analyze the results and present the evolution of the North Rift extent as captured by the InSAR-based delineation. Furthermore, we illustrate the gain of information provided by SAR interferometry versus SAR backscatter and optical images and we attempt to identify the strain variations using double difference interferograms. Finally, Section 7 summarizes the potential, the benefits and the limitations of the proposed method.

**2 Test site**

Brunt Ice Shelf is located along the Caird Coast in East Antarctica, in the eastern sector of the Weddell Sea (Figure 1(a)). It is connected to a larger ice shelf that is made up of the Stancomb–Wills Ice Tongue and the Riiser–Larsen Ice Shelf. The

border with the Stancomb–Wills Ice Tongue is defined at the northeast of BIS by the Brunt–Stancomb Chasm (Anderson et al., 2014).

BIS presents a rich rifting system made of old and new fractures experiencing variable propagation rates, that are described in Thomas (1973) and more recently in De Rydt et al. (2018). The McDonald Ice Rumples (MIR), on the northeast part of the ice shelf tip, originate from the only pinning point constraining the ice flow, and play therefore a key role in the rifting system (Gudmundsson et al., 2017; De Rydt et al., 2019). On the southwest part of BIS, the Chasm 1 rift reactivated in November 2012 and progressively propagated towards the MIR to reach its current extent, after having been dormant for three decades. In October 2016, a new rift named the "Halloween crack" appeared close to the MIR and expanded with a variable speed in the east direction (De Rydt et al., 2018). More resilient than first expected, to date neither Chasm 1 nor the Halloween crack have reached their rupture point yet, although Chasm 1 remains only connected by a short ice bridge of a few kilometers in length.

In November 2020, a third rift - the North Rift - opened at the MIR and quickly spread towards the Brunt–Stancomb Chasm, leading to the calving of iceberg A74 (1270 km²) on 26 February 2021 (British Antarctic Survey, 2021). The formation of this iceberg constitutes the first major calving event on BIS since September 1971 (Thomas, 1973).

## 3 SAR interferometry over ice shelves

The efficiency of repeat-pass SAR interferometry for mapping ice motion and detecting ice surface deformation and strain (e.g. ice flow, grounding lines, grounding of pinning points, etc.) has long been established (e.g. Rignot et al., 1995, 2000, 2011; Mouginot et al., 2019). SAR interferometry is able to capture displacements and deformations of an ice shelf as small as a fraction of the sensor wavelength, making it highly sensitive to changes in comparison with SAR backscatter imagery that can only provide information about cracks at the spatial resolution scale. Since 2014, the Sentinel-1 constellation offers interferometric capabilities at C-band with systematic acquisition, wide coverage, medium resolution and a reduced revisit cycle compared to former missions, therefore providing potential for a regular monitoring of ice shelf rifting activity with interferometry for the first time. The main acquisition mode of Sentinel-1 is the Interferometric Wide swath (IW) mode, based on Terrain Observation with Progressive Scans SAR (TOPSAR) (De Zan and Monti Guarnieri, 2006). In IW mode, SAR images are divided in three swaths, each swath being made of Single Look Complex (SLC) tiles called bursts (Torres et al., 2012). Interferometric processing of Sentinel-1 IW acquisitions requires some specific processing steps, such as deramping and burst stitching, that are explained in Yague-Martinez et al. (2016). The main characteristics of Sentinel-1 SLC products in IW mode are reported in Table 1.

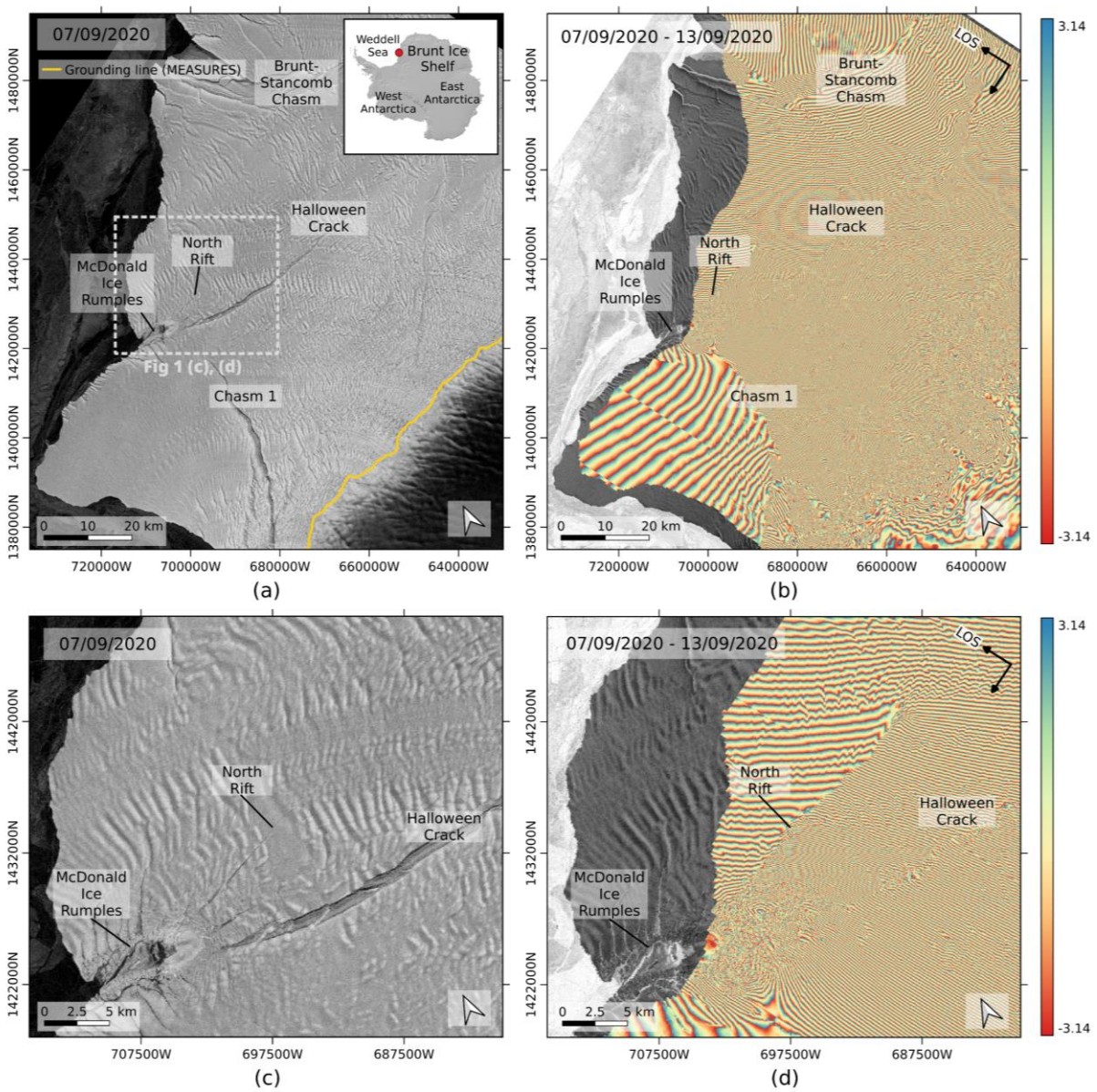

**Figure 1: Rifting system on BIS. (a) Sentinel-1 brightness image acquired on 7 September 2020. The inset indicates the location of BIS over the REMA DEM (Howat et al., 2019) and the yellow line indicates the grounding line provided by the MEaSUREs dataset (Rignot et al., 2011, 2014 and 2016). (b) Sentinel-1 repeat-pass interferogram of 7–13 September 2020. (c) Close up of (a) whose extent is indicated by the gray frame. (d) Close up of (b).**

**Table 1: Sentinel-1 SLC product characteristics in Interferometric Wide Swath mode.**

| Swath Id. | IW1 | IW2 | IW3 |
|---|---|---|---|
| Incidence angle | 32.9° | 38.3° | 43.1° |
| Slant range resolution | 2.7 m | 3.1 m | 3.5 m |
| Azimuth resolution | 22.5 m | 22.7 m | 22.6 m |
| Wavelength | 5.547 cm | | |
| Frequency | 5.405 GHz | | |
| Polarization | HH / VV / HH + HV / VV + VH | | |
| Slant range pixel spacing | 2.3 m | | |
| Azimuth pixel spacing | 14.1 m | | |
| Orbital repeat cycle | 6 / 12 days | | |

## 3.1 Repeat-pass interferometry

Let us consider a repeat-pass interferogram of an ice shelf computed with a master image acquired at epoch $t^i$ and a slave image acquired at epoch $t^j$. Assuming a simple ice shelf model, the phase in the interferogram can be expressed, after subtraction of the flat earth and topographic phase, as a sum of the following components:

$$\phi^{ij} = \phi_{flow}^{ij} + \phi_{tides}^{ij} + \phi_{noise}^{ij}, \tag{1}$$

where $\phi_{flow}^{ij}$ is the ice flow motion phase component, $\phi_{tides}^{ij}$ is the tidal phase component and $\phi_{noise}^{ij}$ is the random phase noise. Any deformation or displacement that does not originate from ice flow or tides is neglected in this simple model. In the following, for the sake of simplicity, we assume that the phase noise is negligible. While the tidal component is determined by the change of the vertical position $D_{tides}^{ij}$ (positive for upward motion) of the ice shelf between epochs $t^i$

and $t^j$ resulting from the balance between oceanic tides and atmospheric pressure (Padman et al., 2003), the ice flow motion phase component is proportional to the surface ice velocity vector $\vec{v}^{ij}$ projected on the line-of-sight (LOS) $v_{los}^{ij}$ and the temporal baseline $\Delta t^{ij} = t^j - t^i$ of the interferometric pair. Each component is exemplified by a sketch in Fig. 2. According to the situations illustrated in Fig. 2, the different phase components can be expressed as:

$$\phi_{flow}^{ij} = \frac{4\pi}{\lambda} v_{los}^{ij} \Delta t^{ij}, \tag{2}$$

$$\phi_{tides}^{ij} = -\frac{4\pi}{\lambda} D_{tides}^{ij} \cos\theta, \text{ and} \tag{3}$$

where $\theta$ is the local incidence angle. The interferometric phase quality is controlled by the coherence, which is a measure of the fringe visibility in the interferogram. In the particular case of repeat-pass interferometry on ice shelves, the

interferometric phase quality can be significantly degraded by temporal decorrelation, e.g. in the event of surface melt, snow fall or wind drift, sometimes leading to a complete loss of information.

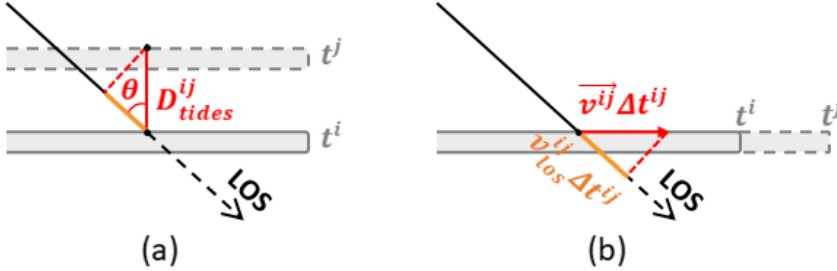

**Figure 2: Displacements contributing to the interferometric phase over an ice shelf. (a) Change of vertical position caused by tides. (b) Horizontal ice flow motion. The ice shelf is represented at times $t^i$ and $t^j$ respectively by the continuous and dashed gray lines.**

In practice, interferograms over an ice shelf exhibit a clear segmentation of the fringe pattern, with discontinuities corresponding to the rifting system. This has already been observed in several cases for different ice shelves – e.g. Brunt, Larsen-C, or Ronne ice shelves (Rignot and MacAyeal, 1998; Larour et al., 2004; Hogg and Gudmundsson, 2017; De Rydt et al., 2018) and it is further illustrated in this paper for Brunt Ice Shelf. In Fig. 1, the rifting system of BIS is pictured and compared to a repeat-pass interferogram generated from Sentinel-1 images acquired 6 days apart in September. We observe that active cracks, rifts and chasms correspond to phase discontinuities in the fringe pattern, dividing the ice shelf into regions with different fringe rates and fringe orientations. As highlighted by Eq. (1), the segmented fringe pattern suggests that SAR interferometry captures the cumulative effect of the ice flow velocity and tidal response, seen as a spatially discontinuous strain field created by rifting activity.

Small-scale moving features such as crevasses may also appear as phase discontinuities because their non-stationary surface roughness cannot be adequately captured by digital elevation models (DEM), thus leaving local residues of topographic phase in the flattened interferogram. However, contrary to rifts and cracks, crevasses do not necessarily concur with the segmentation into regions of changing fringe rate and fringe orientation.

In the region of BIS, the vertical position of the floating ice, as inferred from oceanic tides (CATS2008; Erofeeva et al., 2019) and atmospheric pressure (ERA-5; C3S, 2017) models, may vary by more than 1 m between acquisitions 6 days apart. However, seaward away from the grounding line, the tilt of the ice shelf surface varies little and the change of vertical position for a given pair of acquisitions shows smooth spatial variations of only a few centimeters. In comparison, the ice flow can reach velocities up to 2.5 m d$^{-1}$ (ENVEO CryoPortal), partially captured by the LOS orientation. As a consequence, the tidal phase component corresponds to largely spaced fringes, and the phase signal is dominated by the ice flow motion contribution.

The segmentation of the interferogram into regions with distinct phase ramps related to the rifting activity constitutes the basis of the method presented in the following for delineating cracks automatically. In practice, because the individual phase

contributions cannot be easily discriminated in the interferogram, we assume that both phase components may have a spatially discontinuous behaviour. The poor understanding of the mechanisms driving crack propagation also supports this assumption.

## 3.2 Double difference interferometry

Double difference interferograms, computed as the difference between repeat-pass interferograms, may provide valuable
insights regarding the non-stationary part of the phase signal. Though not used for the crack delineation procedure, double difference interferograms are used in this study to understand the processes at play during the crack propagation.

To highlight the temporally-variable phase contribution of each component, Eqs. (2) and (3) can be reformulated in terms of the variation with respect to the time-invariant core contributions of the ice flow velocity $v_{los}^{core}$ and tidal vertical displacement $D_{tides}^{core}$:

$$\phi_{flow}^{ij} = \frac{4\pi}{\lambda}\left(v_{los}^{core} + \Delta v_{los}^{ij}\right)\Delta t^{ij}, \text{ and} \qquad (4)$$

$$\phi_{tides}^{ij} = -\frac{4\pi}{\lambda}\left(D_{tides}^{core} + \Delta D_{tides}^{ij}\right)\cos\theta. \qquad (5)$$

A double difference interferogram computed from two repeat-pass interferograms spanning respectively epochs $t^i, t^j$ and $t^m, t^n$ has a phase $\Delta\phi^{ij,mn}$ written as:

$$\Delta\phi^{ij,mn} = \phi^{mn} - \phi^{ij}. \qquad (6)$$

Given the almost exact repeat orbit of Sentinel-1, the line-of-sight direction and the incidence angle can be assumed to be constant for all interferograms along a given track. The phase difference can therefore be calculated easily using repeat-pass interferograms geocoded on a common grid. If both repeat-pass interferograms have the same temporal baseline, i.e. $\Delta t^{ij} = \Delta t^{mn} = \Delta t$, inserting Eqs. (4) and (5) into Eq. (6), the phase of the double difference interferogram can be formulated as:

$$\Delta\phi^{ij,mn} = \frac{4\pi}{\lambda}\left[\left(\Delta v_{los}^{mn} - \Delta v_{los}^{ij}\right)\Delta t - \left(\Delta D_{tides}^{mn} - \Delta D_{tides}^{ij}\right)\cos\theta\right]. \qquad (7)$$

Equation (7) shows that the difference between two interferograms with the same temporal baseline removes the time-invariant contribution of the ice flow velocity and the oceanic tides. In the absence of ice flow speedup and other deformations than the tidal ones, the double difference interferogram contains only the change of tidal bending between 4 epochs (3 epochs in case of a common date) (Rignot and MacAyeal, 1998). Nevertheless, if the ice shelf undergoes changes in flow velocity, such differencing would remove the core contributions of both the ice flow and of the tidal deformation and
leave the blend contributions of the differential vertical tidal displacement and the time-variable horizontal ice velocity (acceleration/deceleration) between the repeat-pass interferograms.

Let us note that, in the following, we will refer to repeat-pass interferograms whose flat earth and topographic phase components have been subtracted as *flattened interferograms*, or simply *repeat-pass interferograms*. The terminology

*double difference interferogram* will be used for interferograms computed as the difference of two repeat-pass interferograms.

## 4 Method

To achieve an automatic delineation of active cracks on ice shelves, we propose a method that aims at detecting discontinuities in the phase image. The method consists of four major steps: 1) the generation of a repeat-pass flattened interferogram, 2) the derivation of the phase gradient map, 3) a Canny edge detection applied to the image of the phase gradient magnitude, and 4) the vectorization and cleaning of the edge detection results. The processing line is presented in Fig. 3. The flowchart describes the input data, the auxiliary data, the processing steps and the output product of each step. In the following, each processing step is described in detail and exemplified using the Sentinel-1 interferogram of BIS shown in Fig. 1. The intermediate products corresponding to this example are provided in Fig. 4 and 5, together with close-up views of the North Rift.

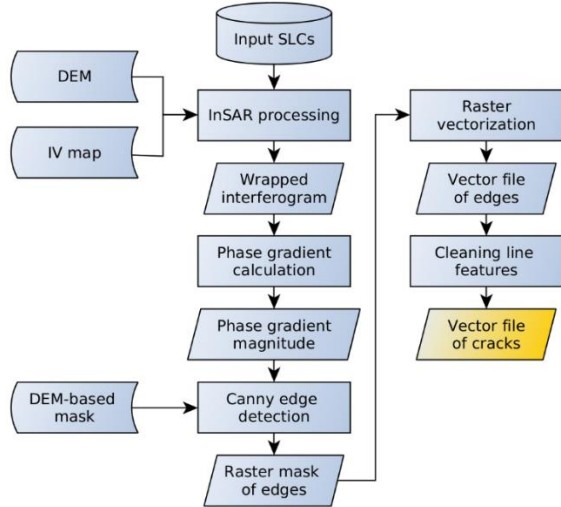

**Figure 3: Processing line for automatic delineation of cracks with InSAR.**

First, Sentinel-1 repeat-pass interferograms are generated from IW SLC acquisitions according to the method presented in Andersen et al. (2020), which is optimized for ice velocity measurements with TOPSAR interferometry. Due to the steering of the antenna from the aft to the fore during each burst acquisition, introducing different viewing angles at the overlap of one burst and the next, Sentinel-1 repeat-pass TOPSAR interferograms may suffer from phase jumps caused e.g. by along-track ice motion (De Zan et al., 2014). Furthermore, ice flow motion, especially in fast-flowing regions, shifts phase centers of the slave image with respect to their location in the master image, leading to potential decorrelation. In addition to accounting for the precise state vectors, the coregistration procedure compensates for the local shifts between the master and slave burst SLCs caused by the along-track and across-track ice flow components, derived e.g. from offset-tracking, in order

to reduce phase jumps and improve the coherence. After coregistration of the master and slave burst SLCs, the interferogram is generated at the burst level, the flat earth and topographic phase components are subtracted and the burst interferograms are stitched together to form an area-wide flattened interferogram.

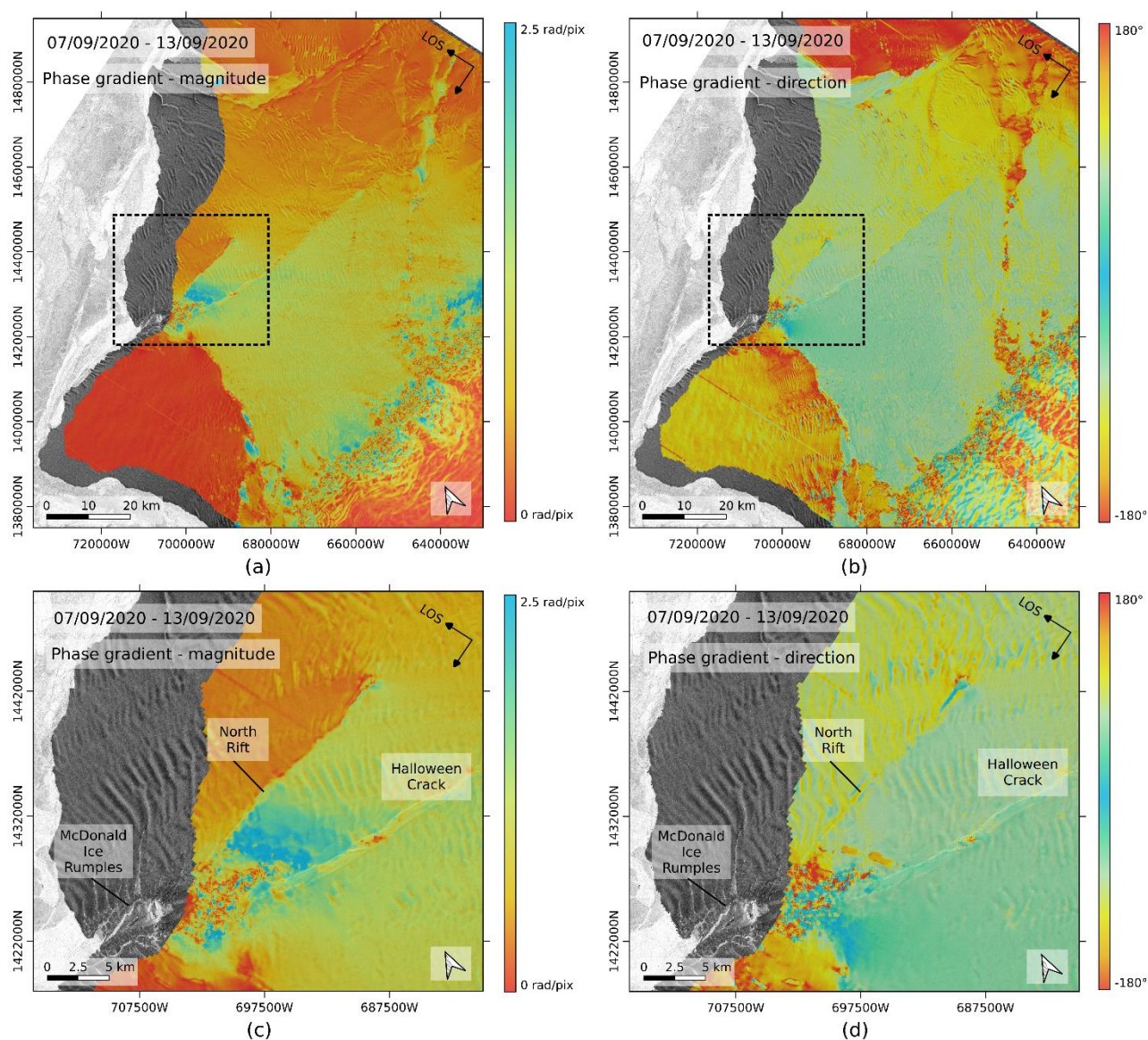

**Figure 4: Phase gradient of the interferogram of 7–13 September 2020 shown in Fig. 1(b). (a) magnitude of the phase gradient. (b)**
**direction of the phase gradient, with angles calculated positive counterclockwise with respect to the horizontal axis. (c) Close-up view of (a), whose extent is indicated by the black dashed frame. (d) Close-up view of (b).**

The phase signal in the interferogram is a sum of the ice motion component, the tidal component and the random phase noise, that results in distinct phase ramps throughout the ice shelf corresponding to separate regions of the rifting system

(Figure 1). The wrapped flattened interferogram is geocoded and subsequent processing steps are applied to the geocoded

fringes.

In the second step, the phase gradient is calculated pixelwise for each geocoded repeat-pass interferogram. Given $\phi_{k,l}$ the value of the wrapped phase for a pixel with coordinates $(x_k, y_l)$ in the interferogram, $k$ and $l$ being the discretization indices in the x- and y-directions respectively, the phase gradient is written as

$$\vec{\nabla}\phi(x_k, y_l) = \left(\frac{\partial \phi}{\partial x}, \frac{\partial \phi}{\partial y}\right)\Big|_{(x_k, y_l)}. \tag{8}$$

The temporal indices $i$ and $j$ are here omitted for the sake of readability. The x- and y-directions refer to the axes of the map projection, which is the Antarctic Polar Stereographic projection (EPSG 3031) in this case. The discrete phase derivatives are computed by averaging the phase differences between adjacent pixels along the x- and y-directions over a square window of $w \times w$ pixels, with $w$ being odd. The averaging is performed using the complex representation of the phase, as expressed by

$$\frac{\partial \phi}{\partial x}\Big|_{(x_k, y_l)} \simeq \angle \left[ \sum_{m=k-\frac{w-1}{2}}^{k+\frac{w-1}{2}} \sum_{n=l-\frac{w-1}{2}}^{l+\frac{w-1}{2}} e^{i(\phi_{m+1,n} - \phi_{m,n})} \right], \text{ and} \tag{9}$$

$$\frac{\partial \phi}{\partial y}\Big|_{(x_k, y_l)} \simeq \angle \left[ \sum_{m=k-\frac{w-1}{2}}^{k+\frac{w-1}{2}} \sum_{n=l-\frac{w-1}{2}}^{l+\frac{w-1}{2}} e^{i(\phi_{m,n+1} - \phi_{m,n})} \right], \tag{10}$$

where $\angle$ represents the argument of the complex exponential. Equations (9) and (10) provide a phase variation per pixel, assuming that pixels are square. If the aspect ratio of the pixel is different than one, then a scaling factor should be applied.

The calculation of the phase gradient translates the complex information provided by the spatially variable fringe pattern into

a two layer real image (i.e. the x- and y- components corresponding to the projection axes in the cartesian case, the gradient magnitude and angle in the polar case). Moreover, by computing the phase gradient directly from the wrapped phase (expressed as a complex number), the tedious step of phase unwrapping is avoided, as well as the phase artifacts that it may introduce.

In practice, the phase gradient is converted to a polar vector, whose magnitude holds the information about the local fringe

rate and whose angle indicates the direction of the phase ramp. For the demonstration case, the images of phase gradient magnitude and angle are shown respectively in Fig. 4(a) and 4(b). In both the magnitude and direction images, the location of the phase discontinuities, i.e. the active cracks, is enhanced and has become easy to identify (e.g. North Rift, Chasm 1 or Brunt–Stancomb Chasm). For most of the identified rift structures, the edges in the magnitude and direction images indicate similar locations. However, for wide open chasms with a complex structure (e.g. the widest part of Chasm 1 or the Brunt–

Stancomb Chasm), the magnitude and direction of the phase gradient may picture a slightly different fractured area. Given that the crack locations mapped by both indicators are mostly similar and that the angles are wrapped by nature, which

makes them difficult to manipulate, we neglect the phase gradient direction and focus on the information held by the phase gradient magnitude.

In the phase gradient magnitude image, active cracks correspond generally to a well-defined step-edge with variable contrast. Crack delineation is hence performed by applying a Canny edge detection to this image (Canny, 1986). In order to reduce the noise while preserving the edges, a median filter is applied beforehand. In practice, the Canny edge detection consists of computing the intensity gradient of the input image and applying a double threshold for mapping the edges: the upper threshold discriminates the strong edges; the lower threshold is used for selecting the weak edges, meant to connect the strong edges present in their neighborhood. An additional Gaussian filtering, with tunable standard deviation, is performed as part of the Canny edge detection procedure for reducing the noise before computing the intensity gradient.

We focus on the rifting activity on the ice shelf and therefore mask the areas of grounded ice. In our case, the masking is performed with a simple thresholding of the TanDEM-X global digital elevation model at a 50 m height, that shows a rough agreement with the grounding line location on BIS. Patchy decorrelation can also cause erroneous edge detection and areas with low coherence (< 0.12) are therefore also excluded.

As shown by the blue lines in Fig. 5, applying the Canny edge detection to the gradient magnitude efficiently maps the cracks present on BIS. However, it also maps small "dangles" (loose curvy lines) caused by phase artifacts, topography or crevasses. As a final step, the raster mask generated by the Canny edge detection module is thinned, vectorized and cleaned using GRASS GIS tools. The cleaning consists of removing small dangles, as they are less likely to correspond to a major propagating crack. The cleaning step is critical and the result should be carefully evaluated, because poor thresholds can lead to a substantial loss of detectable cracks. Though necessary for removing noise and obtaining readable information, the cleaning process sets a bound on the minimum size of detectable cracks, depending on the dangle size threshold.

As shown by the red lines in Fig. 5, the noisy aspect is reduced after the cleaning step. Some residual errors remain, especially in the region near the grounding line. This area is highly crevassed due to the tidal bending and the rapid height change at the transition between floating and grounded ice. These residual dangles thus correspond to a damaged area, though crevasses are not the damage that we aim at mapping. Future studies might investigate the density of detected edges in the vicinity of the grounding line as an indication of the degree of crevasse damage.

It is worth noting that the edge detection thresholds are dependent on the time interval between the acquisitions, the viewing geometry and the spatial variation of the strain rates introduced by the rifting activity. This can be easily shown for the 1-D case. We assume, as discussed in Section 3, that the main contribution to the interferometric phase comes from the ice flow motion: $\phi^{ij} \cong \phi^{ij}_{flow}$. Given this assumption, the phase gradient magnitude can be expressed using Eq. (2) for the 1-D case as:

$$\left|\frac{\partial \phi^{ij}}{\partial x}\right| \cong \left|\frac{4\pi}{\lambda}\Delta t^{ij}\frac{\partial v^{ij}_{los}}{\partial x}\right| = \left|\frac{4\pi}{\lambda}\Delta t^{ij}\cos\theta\right|\left|\frac{\partial v^{ij}}{\partial x}\right|, \tag{11}$$

The Canny edge detection performs a double-thresholding on the gradient magnitude of the input image, which is the phase gradient magnitude described by Eq. (11). The thresholds are hence applied to the gradient magnitude of the phase gradient magnitude, that is described by:

$$\left|\frac{\partial}{\partial x}\left|\frac{\partial \phi^{ij}}{\partial x}\right|\right| \cong \left|\frac{4\pi}{\lambda}\Delta t^{ij}\cos\theta\right|\left|\frac{\partial}{\partial x}\left|\frac{\partial v^{ij}}{\partial x}\right|\right|. \tag{12}$$

The gradient intensity described by Eq. (12) is dependent on the local incidence angle $\theta$ and proportional to the temporal baseline $\Delta t^{ij}$. We also observe a dependence on the absolute spatial variation of $\left|\frac{\partial v^{ij}}{\partial x}\right|$, the term $\frac{\partial v^{ij}}{\partial x}$ being a component of the strain rate tensor (Alley et al., 2018). A first approximation of the edge detection thresholds can therefore be obtained if the strain rate variations caused by the rift propagation can be estimated: the lower threshold should discriminate the smooth natural strain rate variations of the ice sheet background; the upper threshold should be defined by the minimum value of the strain rate variations that is certainly associated to rifting.

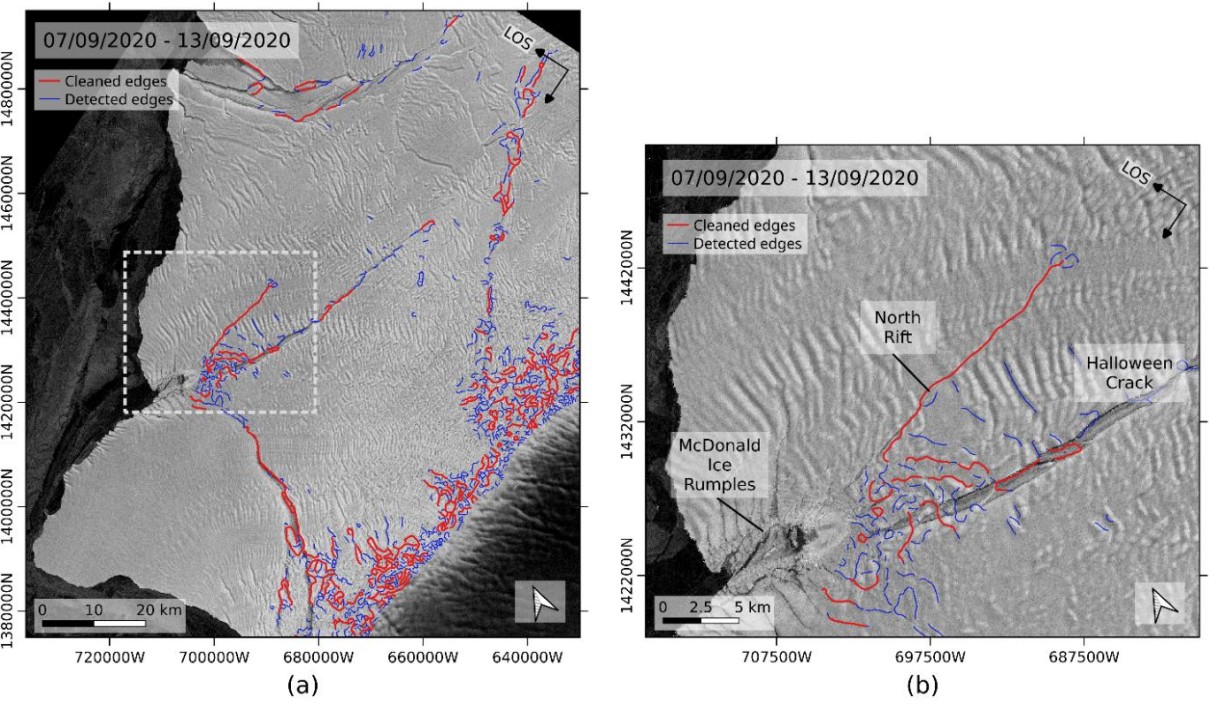

**Figure 5: Cracks automatically delineated from the interferogram of 7–13 September 2020. (a) Edge detection results, before (blue) and after cleaning (red) of the small dangles. The Canny edge detection applied to the phase gradient magnitude shown in Fig. 4. (b) Close-up view of (a), whose extent is indicated by the gray dashed frame.**

Although the InSAR-based crack delineation performs well under conditions that preserve the phase coherence, its applicability is primarily limited by the quality of the SAR interferogram. In case of fast-flowing ice, snowfall, surface melt or snow drift caused by katabatic winds, the InSAR signal decorrelates and the method cannot be applied. Even so, the

regularity of Sentinel-1 acquisitions offers an increased likelihood of coherent interferometric pairs. Another limitation directly originates from TOPSAR interferometric processing, which is strongly affected by coregistration errors and uncorrected ionospheric delays, that may leave residual phase discontinuities at the burst overlap (e.g. see the western tip of the ice shelf in Fig.1(b)), hence causing potential false detections (De Zan et al., 2014).

Let us note that, in the illustrating example, the Halloween crack is only partially mapped even though it is visible in the
brightness images. The fringe pattern is similar on both sides of the crack and the corresponding gradient discontinuity appears faint in the gradient image, which might indicate that the rift was not propagating during the investigated time period.

## 5 Dataset and processing

To capture the propagation of the North Rift and the calving of A74, the InSAR-based method for crack delineation is tested
on a dataset of Sentinel-1 HH-polarized SLC images acquired every 6 days between 1 September 2020 and 6 March 2021, along track 50 (Figure 6). We selected the frames covering BIS and generated all available 6-day repeat-pass interferograms. Overall, 32 interferograms were generated, out of which 13 could not be used for crack delineation because of signal decorrelation. In particular, the interferograms spanning the periods directly before and after the calving event (i.e. between 10 February 2021 and 6 March 2021) could not be used for mapping the North Rift with InSAR. Because the amount of
fringes caused by ice motion increases with the temporal baseline, most of the Sentinel-1 interferometric pairs acquired along track 164, with 12-day repeat pass, were decorrelated. Given their limited quality and temporal resolution, these pairs are not used for crack delineation, but only to support the interpretation and analysis of the temporal evolution of fracturing.

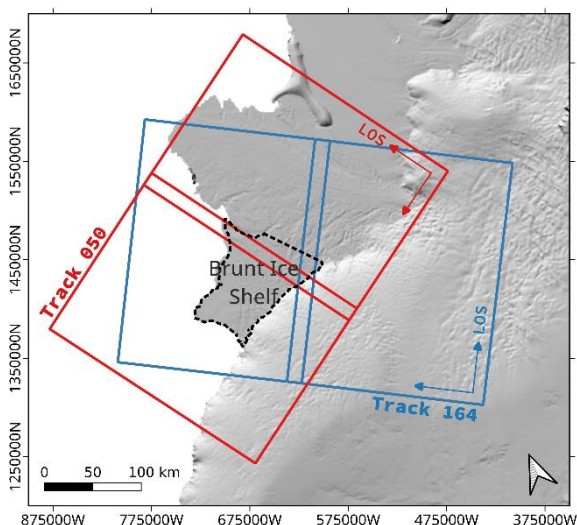

**Figure 6: Sentinel-1 coverage of BIS: track 50 (red) and track 164 (blue). The background is a shaded relief of the REMA DEM**
**(Howat et al., 2019). In IW mode, incidence angles vary from about 30° in the near-range to 45° in the far-range.**

In order to reduce the phase noise, the interferograms are computed with a multilooking factor of $9 \times 3$ pixels in the slant range and azimuth directions, respectively, and an adaptive Goldstein filtering is applied for further noise reduction (Goldstein and Werner, 1998; Baran et al., 2003). For the coregistration, the average ice flow motion is compensated using a multiannual ice velocity map with 200 m pixel spacing generated within the ESA Antarctic Ice Sheet CCI project (ESA Climate Office: Ice Sheets Antarctic). The Antarctic velocity map is calculated using offset-tracking applied to all Sentinel-1 6- and 12-day repeat pairs available for the mission lifetime (2014–today). The offset-tracking processing is described in Nagler et al. (2015), and an additional correction of the vertical displacement induced on floating ice by differential tides (CATS2008; Erofeeva et al., 2019) and atmospheric pressure (ERA-5; C3S, 2017) is applied to offset-tracking results. The topographic phase is estimated and subtracted from the repeat-pass interferograms using the TanDEM-X polar DEM with 90 m grid resolution, extended to cover ice shelves (Wessel et al., 2021). Finally, the interferograms are geocoded on a grid with a 40 m pixel spacing in the Antarctic Polar Stereographic reference system, that matches approximately the pixel size in the radar geometry.

For each geocoded repeat-pass interferogram, the phase gradient is calculated over a window of $w \times w = 9 \times 9$ pixels and the image of the phase gradient magnitude is further filtered using a median kernel of $9 \times 9$ pixels. The Gaussian kernel has a standard deviation equal to 5, and the lower and upper thresholds for edge detection are set respectively to 0.15 and 0.21. Furthermore, all areas above 50 m height or having a coherence lower than 0.12 are masked out for the edge detection procedure. The raster output of the edge detection is thinned, vectorized and cleaned by applying a threshold of 2000 m on the dangle size.

The edge detection parameters are determined for this particular test case by testing different sets of values on a single pair of acquisitions, and fine-tuned to achieve a balance between detections, segment connectivity and false alarms. Considering the IW2 case with a 6-day repeat interval (see Table 1), the chosen upper and lower thresholds correspond respectively to strain rate local variations of about $1.94 \ 10^{-4} \ d^{-1}$ and $1.39 \ 10^{-4} \ d^{-1}$. For other test sites, the detection thresholds might need to be adapted according to the repeat cycle of acquisitions and the viewing geometry with respect to the velocity field.

## 6 Results and discussion

### 6.1 Interferogram time series

The sensitivity of Sentinel-1 repeat-pass interferograms to the North Rift propagation is illustrated by a time series of interferograms presented in Fig. 7. These repeat-pass interferograms are all acquired along track 50 and computed as described in the previous section. The black dashed line represents the manually delineated calving front location, after the calving event of A74, and it is used to picture the North Rift's maximum extent. In this figure, it is visible that the discontinuity line expands along the crack location from one interferogram to the other. The magnitude and direction of the phase gradient is highly variable and provides a first qualitative indication of the propagation timeline of the North Rift: in November 2020, the rift propagates as a straight line in a given direction and the phase ramp on the northern side of the rift

varies gently; at the end of December, after a complete rotation of the fringe pattern, the crack changed direction to propagate toward the Brunt–Stancomb Chasm; at the end of January, when part of the northern plate is decorrelated and the fringe pattern is very tight, it is visible that the crack almost reached the chasm.

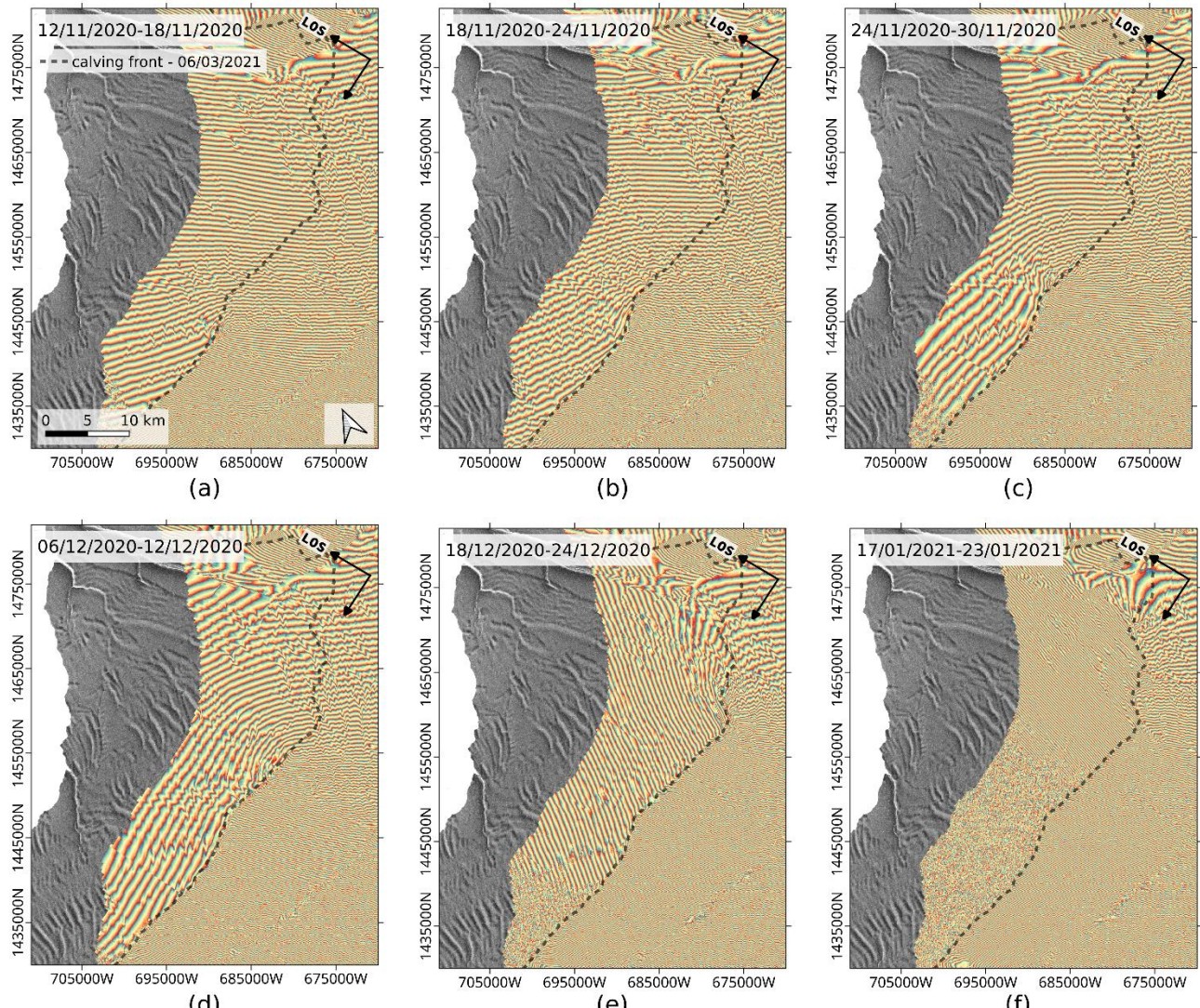

**Figure 7: Time series of wrapped repeat-pass interferograms showing the phase ramp variation from one date to another, as the North Rift propagates between November 2020 and January 2021. The black dashed line indicates the full extent of the North Rift, manually delineated from the calving front on 6 March 2021, after the A74 formed.**

## 6.2 Automatic delineation of cracks

The proposed InSAR-based method for crack delineation is applied as described in the previous section to produce a time series of vector files mapping the BIS crack system with a 6-day resolution, only interrupted whenever the interferometric phase is decorrelated. Though a rough estimate could already be obtained from the visual analysis of the interferogram time

series, the automatically delineated cracks enable a refined and more objective determination of the North Rift propagation from the start of the crack propagation until the calving of A74. In Fig. 8, some selected repeat-pass interferograms are displayed at a larger scale together with the phase gradient vector field and the automatically delineated cracks. It is seen that the North Rift started to open up a few kilometers between 18 and 24 November 2020. Indeed, the phase discontinuity indicated by the yellow line expands between 12–18 November 2020 and 18–24 November 2020. Around the expanding end of the crack, we observe a rotation of the phase gradient that corresponds to a change in the strain pattern. Compared to the November interferograms, the interferogram of 18–24 December 2020 shows a 1.5 to 2.5-fold intensification of the phase ramp around the rift and a change in direction. At this date, the crack already changed direction, after propagating on several tens of kilometers eastward.

The propagation history is summarized in Fig. 9. The left panel shows the automatically delineated cracks for the overall dataset, the pairs with low coherence being excluded. In the right panel, the focus is set on the North Rift and the major steps of its advance are highlighted with a selection of relevant dates. The background is a Sentinel-1 brightness image acquired on 6 March 2021, that shows a good agreement between the delineated cracks and the calving front after the iceberg A74 formed. As observed also in Fig. 8, the crack started to open sometime between the 18 and 24 November 2020, and continued to rapidly propagate in early December. As the rift opened, both sides of the crack slowly drifted away from each other, leading to a massive coherence loss on the northern side of the rift and some erroneous detections after mid-December. Despite the phase noise, the crack delineation method was able to map the easternmost part of the North Rift and the results agree well with the shape of the manually derived calving front. From the interferogram of 17–23 January 2021, about a month before the iceberg broke off, the quasi-full extent of the calving could already be mapped.

### 6.3 Propagation rates

In addition to delineating the North Rift, we also estimate its length for each coherent pair of acquisitions shown in Fig. 9(b). In practice, the crack length is estimated as the cumulative length of the delineated segments corresponding to the North Rift. In the case of incomplete or interrupted segments, e.g. due to decorrelation, the missing segment length is estimated manually. The evolution of the estimated rift length over time is plotted in Fig. 10. We observe that, for most pairs, the North Rift consistently grows over time as expected. However, for a few pairs, the crack length is estimated to be smaller by a few hundred meters than the extent of the prior pair of acquisitions. This inconsistency arises from the coarse accuracy of the estimation method, that is used for analysis purposes rather than for accurate quantitative estimates. Based on the estimated crack extent, the propagation rates are derived as the segment slope between two consecutive points. The propagation rates are reported by the annotations in Fig. 10. The negative propagation rates resulting from an inconsistent decrease of the North Rift length are omitted. The propagation rate is the largest (about 1.3 km d$^{-1}$) when the North Rift activates at the end of November and reaches a similar value at the latest stage of propagation in January. In between, we observe three period of rapid expansion separated by two plateaus showing almost no progression of the rift. During the intermediate propagation period, the progress of the North Rift extent is in the order of several hundred meters per day.

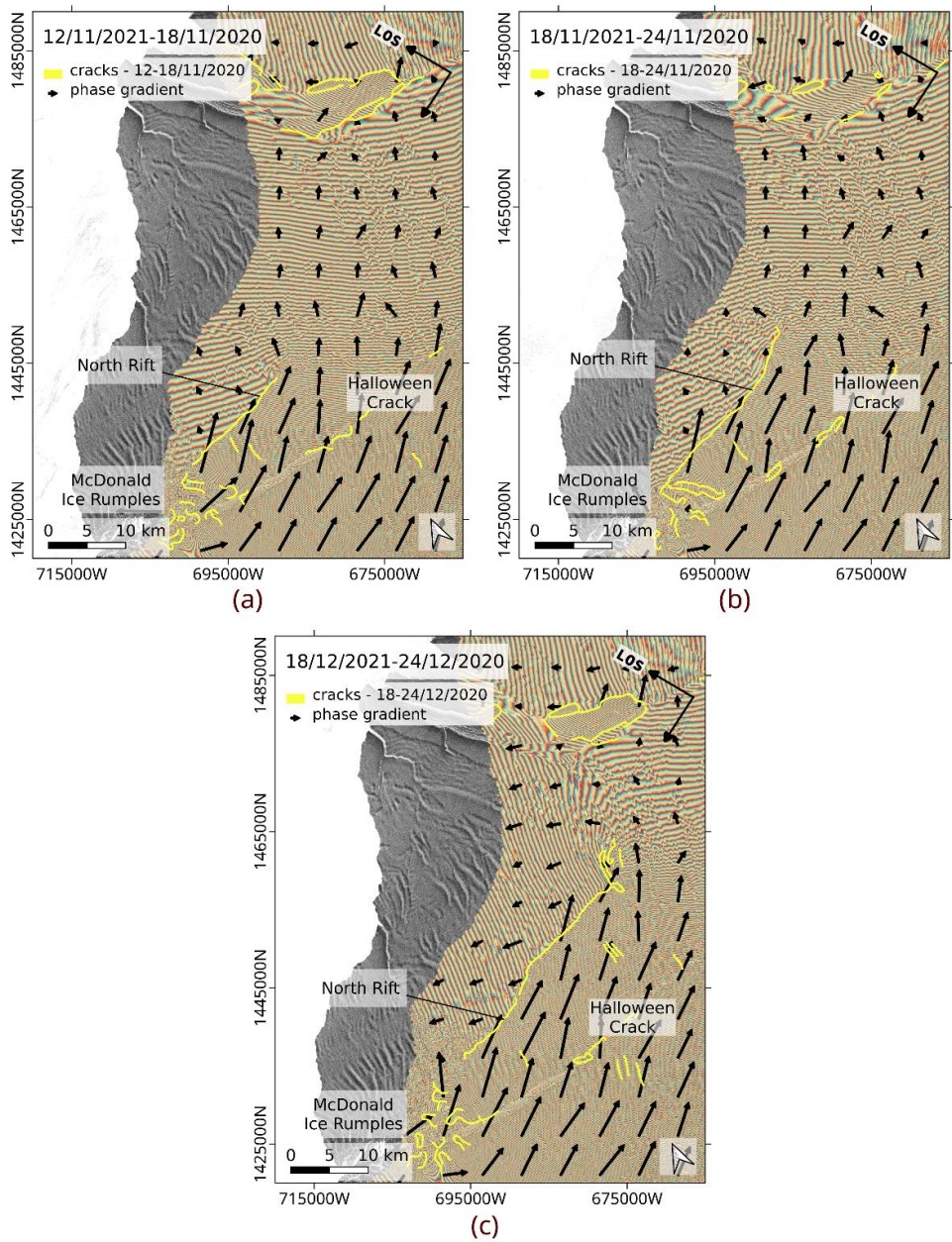

**Figure 8: Sentinel-1 repeat-pass interferograms on BIS, overlaid with the cracks delineated from the phase gradient magnitude (yellow lines). The phase gradient is pictured by the vector field. (a) 12–18 November 2020. (b) 18–24 November 2020. (c) 18–24 December 2020. The comparison of the interferograms shows a rapid propagation of the North Rift between 18 and 24 November 2020. In December 2020, the crack almost reached the Stancomb–Wills Chasm. As the crack propagates, the phase gradient intensifies and changes direction on both sides of the rift.**

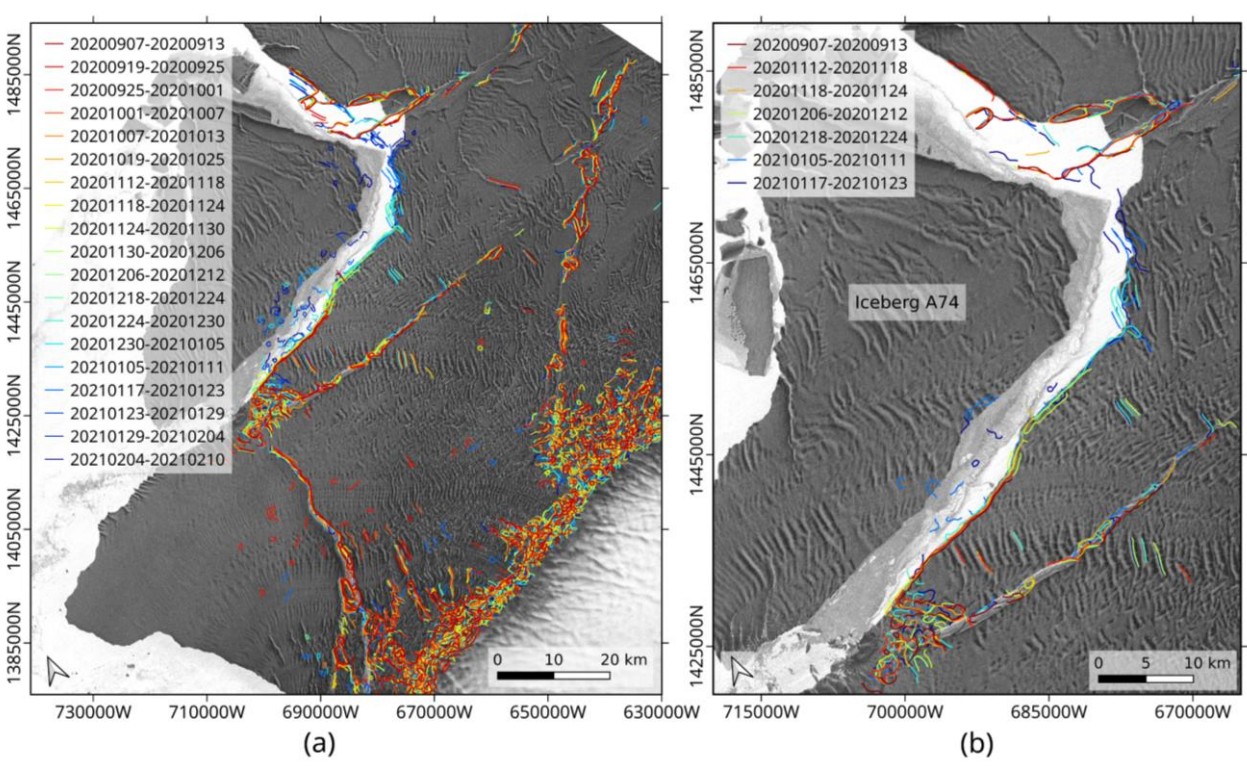

**Figure 9: (a) Evolution of BIS rifting system between 7 September 2020 and 23 January 2021 from automatically delineated cracks. (b) Propagation history of the North Rift with selection of significant dates. The delineated cracks are overlaid on a Sentinel-1 brightness image acquired on 6 March 2021, showing the calving front shape after the iceberg A74 formed.**

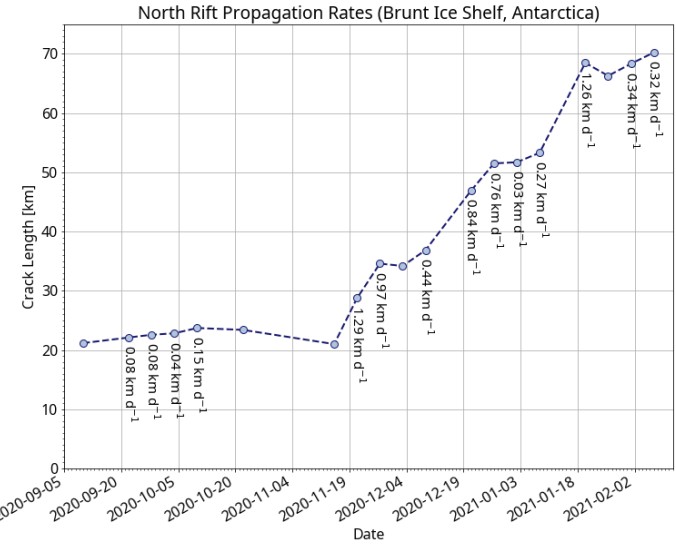

**Figure 10: North Rift propagation rates.**

## 6.4 Comparison with SAR backscatter and optical imagery

In order to stress the added value of SAR interferometry for mapping active rifts, we compare the InSAR-based detections of the North Rift against SAR and optical imagery. Let us first consider the SAR backscatter images. For crack monitoring, the performance of SAR backscatter imagery is mostly limited by the image resolution and the orientation-dependent
backscatter. The contrast between the thin fractures and the ice shelf background is often lacking and further jeopardized the speckle noise. In Fig. 11, we present the evolution of the North Rift as seen with a time series of Sentinel-1 backscatter images. In these brightness images, the North Rift is visible in November, but shows a relatively small extent compared to that mapped with interferometry (see interferograms in Fig. 7). The well-advanced breach is visible in the brightness images only a few days before the calving of the iceberg. Comparatively, the interferogram of 17–23 January 2021, fully captures
the extent of the crack that becomes visible only a month later in the brightness image acquired on 22 February 2021.

When available, optical images may constitute a better option than the SAR backscatter: although their performance for observing fractures is also limited by the image resolution, the ice shelf background appears smoother and it provides a better contrast of the cracks. Though suffering from major caveats such as the dependence on solar illumination and cloud cover that prevent systematic monitoring of ice shelves, optical imagery provides a good opportunity for occasional
observations and as reference dataset. Coincidently with our Sentinel-1 dataset, two Landsat-8 cloud-free images of BIS from the Operational Land Imager (OLI) were acquired on 19 January 2021 and 6 February 2021. The North Rift is well visible in these images and its location can be extracted for validation of the InSAR-based results. For this purpose, we manually delineate the position of the NR in both Landsat-8 images. The NR location derived from each Landsat-8 images is compared with the NR location derived from InSAR using the Sentinel-1 pair of 17–23 January 2021. This Sentinel-1 pair is
coincident with the first Landsat image and it is also the last coherent one in the dataset before the calving event. The comparison, that consists of calculating the shortest distance between the Landsat and InSAR-based NR delineations, is presented in Fig. 12. We observe that, for the Landsat-8 image of 19 January 2021, the agreement is better than 200 m for most sections of the crack, but the InSAR detection pictures the crack longer than the Landsat delineation, especially around the tip of the crack and along secondary branches. The loop at the tip of the InSAR-based detection may appear like a
detection artifact, but we see that this tip agrees well with the curvature of the NR delineated from the Landsat-8 image of 6 February 2021. The better agreement of the tip with the NR observed at a later stage of expansion suggests that the loop, though wrongly connected, is an actual detection and that SAR interferometry detected signs of the rift propagation a few days ahead the optical imagery.

## 6.5 Strain pattern variations

Mapping cracks with SAR interferometry relies on the assumption that interferometric phase captures the dynamic changes caused by rifting activity. The variability of the phase gradient magnitude and orientation as the rift propagates has been illustrated in Fig. 7 and Fig. 8. However, the origin of these temporal variations remains difficult to determine from repeat-

pass interferograms, as the differential phase is a blend of ice flow and intrinsically variable tidal displacements, as shown by Eq. (1).


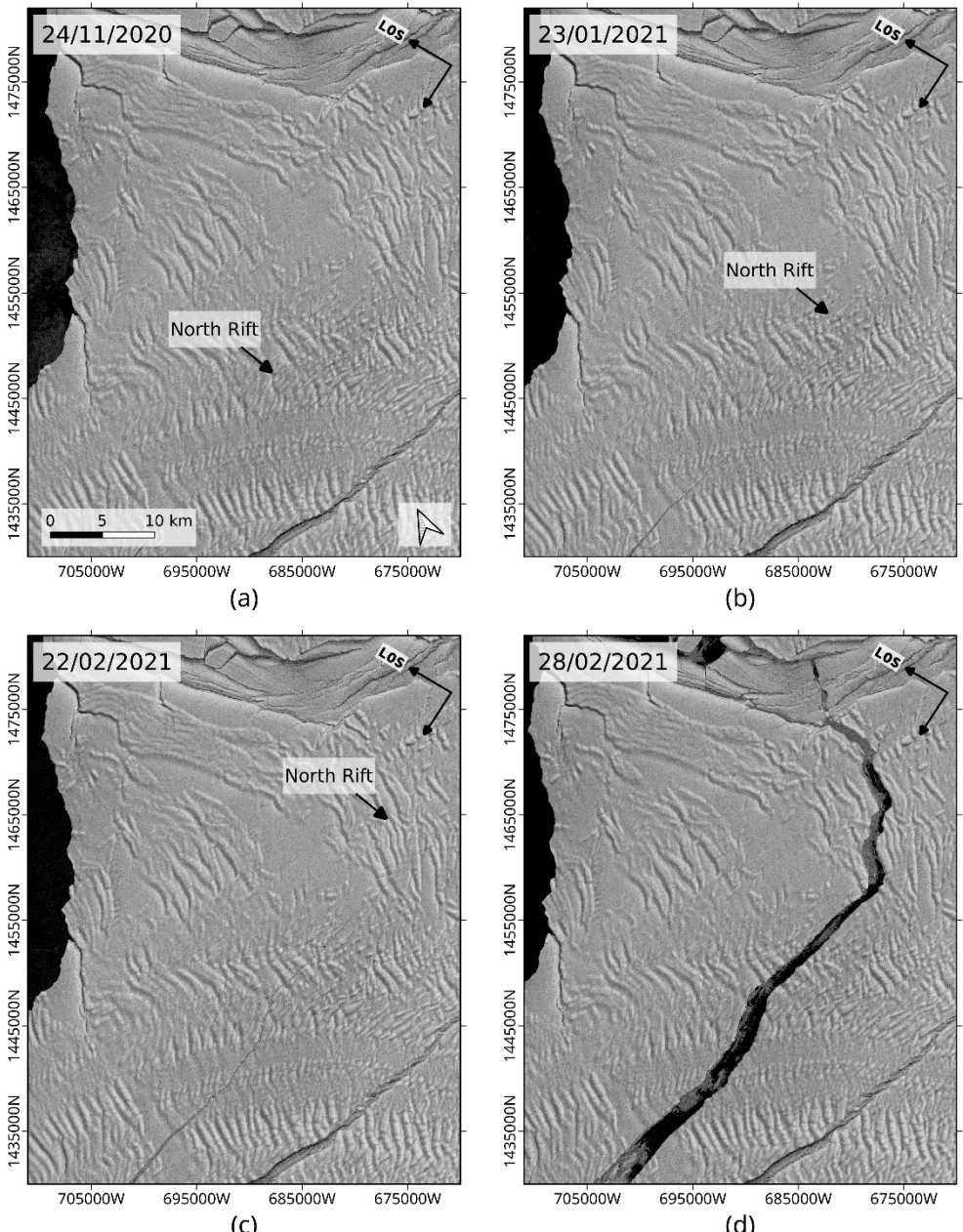

**Figure 11: Propagation of the North Rift between end of November and end of February observed with a series of Sentinel-1 radar brightness images. The iceberg A74 formed on 26 February 2021. The images focus on the same area as the interferograms in Fig. 6. (a) 24 November 2020. (b) 23 January 2021. (c) 22 February 2021. (d) 28 February 2021.**

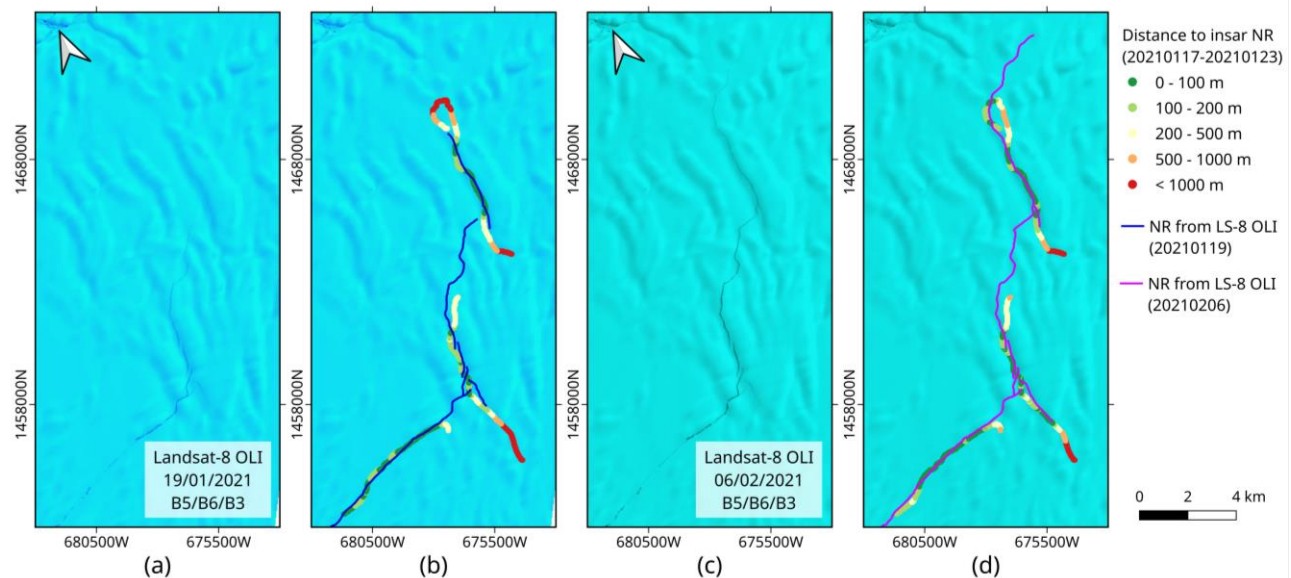


**Figure 12: Tip of the North Rift observed with Landsat-8 OLI RGB composites from bands 5-6-3. The composites are overlaid with the North Rift extent manually delineated from the Landsat image and compared with the S-1 InSAR-based North Rift of 17–23 January 2021, color-coded to represent the distance to the location in the Landsat image. (a) Landsat-8 composite of 19 January 2021. (b) Distance between the North Rift detected with InSAR on 17–23 January 2021 and the North Rift delineated from Landsat-8 on 19 January 2021 (blue). (c) Landsat-8 composite of 6 February 2021. (d) Distance between the North Rift detected with InSAR on 17–23 January 2021 and the North Rift delineated from Landsat-8 on 6 February 2021 (pink).**

In this section, we aim at emphasizing and interpreting the temporally-variable component of the phase signal introduced by rifting activity. For this purpose, we use the rationale presented in Section 3.2 and calculate differences between consecutive 6-day repeat-pass interferograms along track 50. These double difference interferograms are shown in Fig. 13. The first one is the difference between the interferograms of 12–18 November and 18–24 November 2020. It shows curved fringes on the expanding tip of the crack, likely caused by the strain of the diverging ice plates as the crack starts to propagate. In the second double difference interferogram (18–24 November and 24–30 November 2020), the strain becomes almost evenly distributed on both sides of the rift (14 fringes on the north plate and 12 on the south plate). This number of fringes corresponds to a change in line-of-sight displacement of about 35 cm relative to the origin of the crack, the line-of-sight being almost parallel to the fringe orientation. Finally, in December 2020, when the crack propagation changes direction, the fringe pattern becomes very tight and remains almost evenly balanced on the north and south plates.

Without the information from another viewing direction, the observed phase ramp could either correspond to a vertical deformation or to a horizontal displacement/speedup. For solving this ambiguity, we also display the difference between 12-day interferograms acquired in November along track164, which has a line-of-sight rotated by about 60° with respect to track 50. Interferograms along this orbit have a lower temporal resolution and quality, and they are therefore only used here to support the analysis. The double difference results in this case into a homogeneous fringe pattern with fringes oriented nearly parallel to the crack.

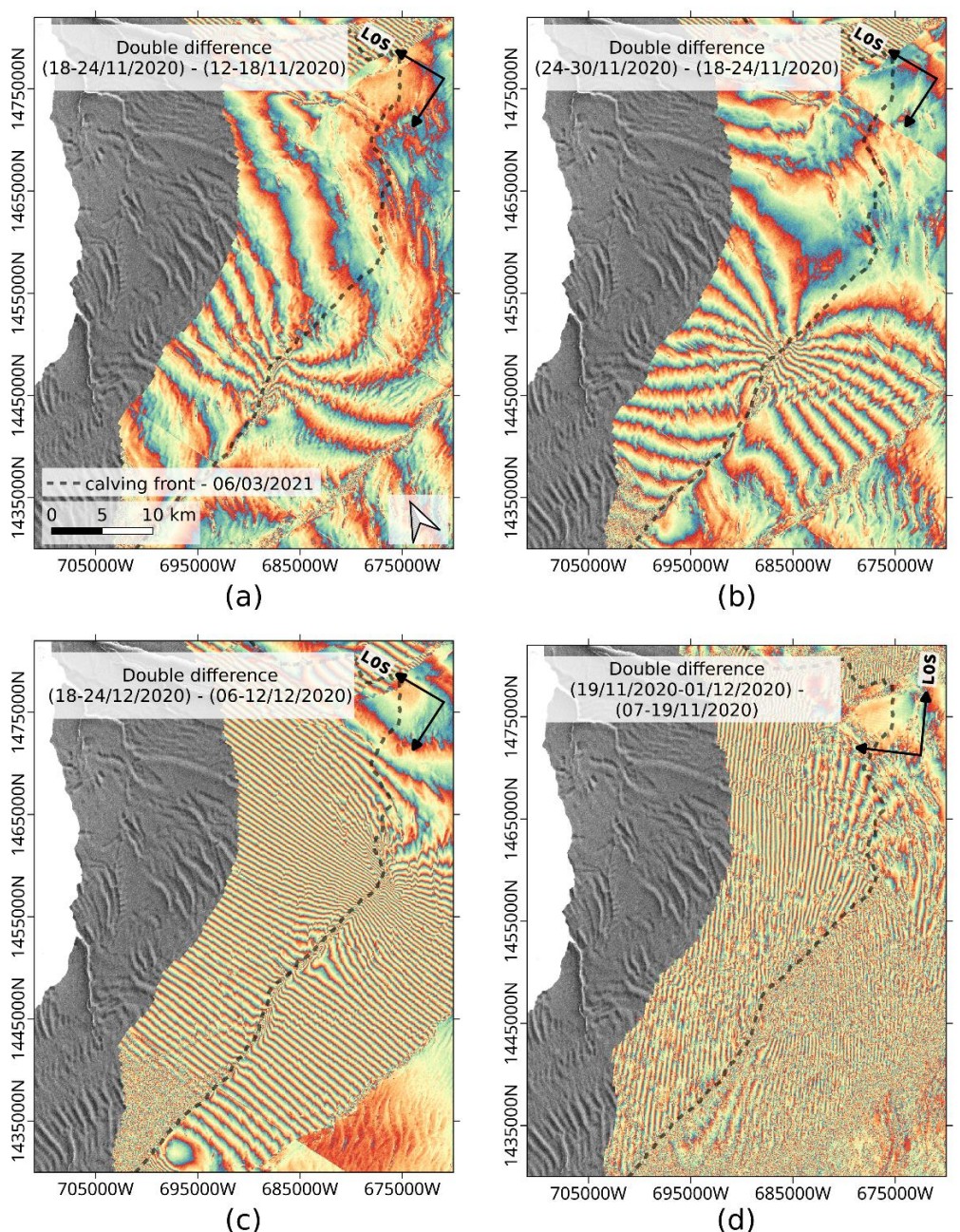

**Figure 13: Difference between consecutive repeat-pass interferograms showing the strain pattern variation as the North Rift propagates. (a)-(c) are calculated from 6-day interferograms of track 50 shown in Fig. 7. (c) is calculated from 12-day interferograms of track 164 (not shown). The black dashed line indicates the full extent of the North Rift. (a) Difference between 18–24 November 2020 and 12–18 November 2020 (track 50). The crack starts to propagate. (b) Difference between 24–30 November 2020 and 18–24 November 2020 (track 50). The crack propagates as a straight line along a given direction. (c) Difference between 18–24 December 2020 and 6–12 December 2020 (track 50). The crack has almost reached the Stancomb–Wills Chasm. (d) Difference between 19 November–1 December 2020 and 7–19 November 2020 (track 164). The crack has reached its diverging point.**

For all three double difference interferograms along track 50, the color order of the fringes is reversed in the regions north and south of the cracks: on the calving front side, the differential phase increases from the expanding tip towards the MIR, while it decreases in the same direction in the region between the NR and the Halloween crack. A positive phase corresponds to a change positive in the direction away from the satellite. Similarly, the double difference interferogram along track 164 shows an increase of the differential phase from the NR towards the calving front, as well as an increase of the phase from the NR towards the Halloween crack.

For both tracks, the double difference interferograms show fringes nearly parallel to the line-of-sight direction around the North Rift. Peltzer et al. (1994) simulated the fringe pattern due to a rigid-body rotation around an axis that is perpendicular to a horizontal surface and demonstrated that it would create such fringes parallel to the viewing direction. Later on, Rignot and MacAyeal (1998) also observed fringes parallel to the LOS direction on Ronne Ice Shelf and interpreted it as rigid-body rotation around an axis perpendicular to the ice shelf surface and located at the tip of the crack. Because they observed this pattern in the repeat-pass interferograms but not in the double difference ones, they attributed this rigid-body rotation to a velocity difference between both sides of the rift, not to a transient phenomenon. In the case of the North Rift, fringes parallel to the line-of-sight are observed with two different viewing directions and a rigid-body rotation around the tip of the crack seems therefore likely. Moreover, the different phase trends on the north and south sides of the North Rift actually suggest that two distinct rigid-body rotations occur, with opposite directions of rotation. Since this pattern is present in the double difference interferograms, it would be in this case associated to a time-varying response to rift propagation that is larger than the differential tidal displacement in the vertical direction. The hypothesis of the rigid body rotation is further strengthened by ice velocity measurements performed with offset-tracking and shown in Fig. 14. The offset-tracking measurements show an acceleration of the ice flow around the southwest part of the North Rift from December 2020 onwards. Simultaneously with the North Rift propagation in November–December, a speedup of about 0.5 m d$^{-1}$ is measured on the tip of the northern plate. In January, the measured velocity further increases to reach a speedup of about 2 m d$^{-1}$ on the tip. The amplitude of the speedup varies from the southwest to the northeast, similarly to the strain pattern observed in the double difference interferograms, and this acceleration comes together with a rotation of the velocity vector field towards the northeast and southwest directions respectively north and south of the rift. The change in orientation of the measured velocity field fits indeed two distinct rotations: one with a clockwise rotation around the tip of the North Rift north of the rift, and one with counterclockwise rotation south of it. North of the rift, the observed rotation pattern could originate from ice flow speedup due to the loss of constraint at the MIR, or it could correspond to the displacement of the future iceberg as it separates progressively from the ice shelf after the opening of the North Rift over the full ice thickness and its consequent widening. In the second case, the displacement would be misinterpreted as ice flow acceleration by offset-tracking.

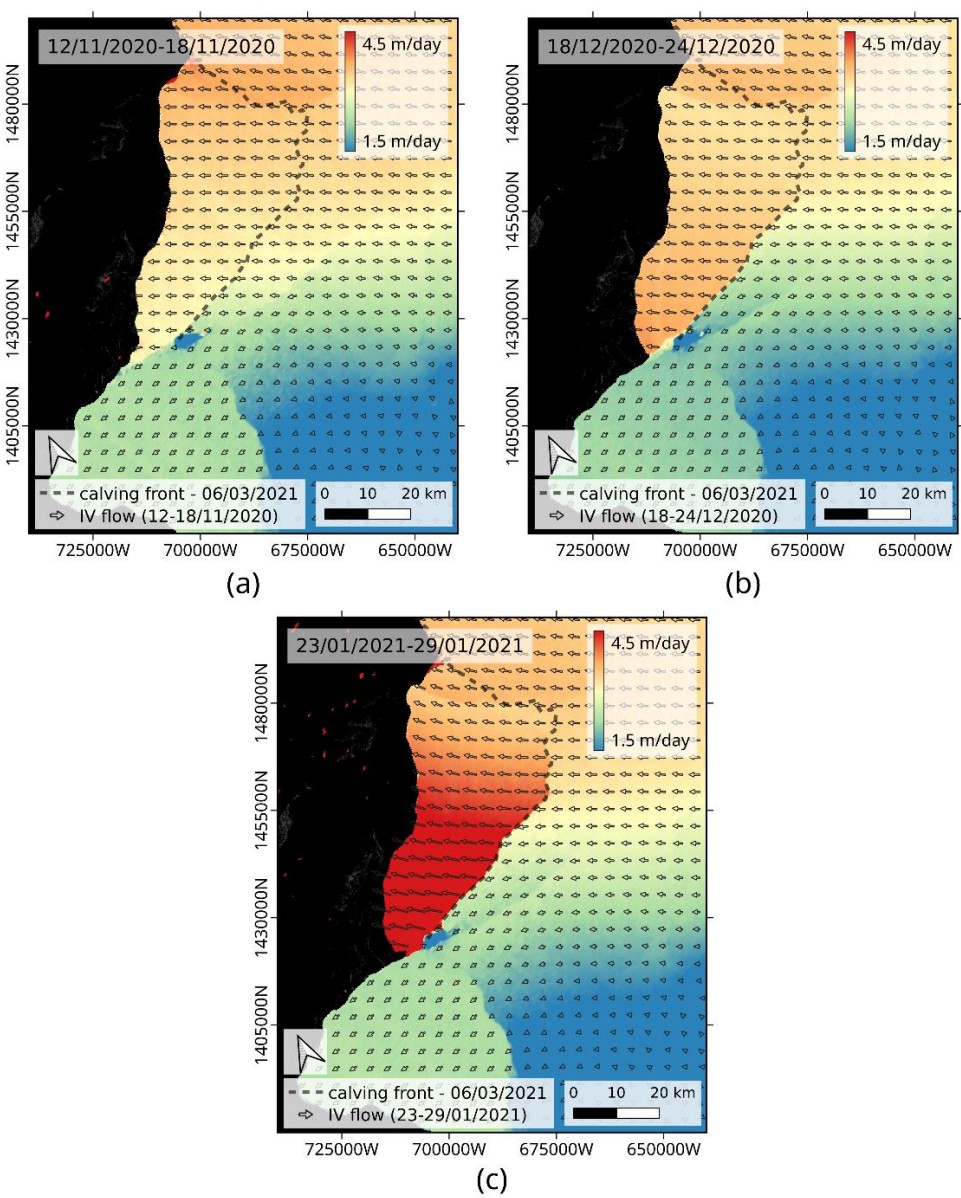

**Figure 14: Ice velocity field over BIS calculated with offset-tracking applied to Sentinel-1 6-day pairs over track 50. The background image is the magnitude of the ice velocity and the overlaid vector field represents the ice flow vector. The results are corrected for vertical tidal displacements using the CATS2008 model for tides and the ERA-5 model for atmospheric pressure. (a) 12–18 November 2020. (b) 18–24 December 2020. (c) 23–19 January 2021.**

530

## 7 Conclusion

The InSAR-based method proposed for automatic crack delineation has been successfully tested and qualitatively validated on BIS with a dataset of Sentinel-1 6-day repeat-pass interferograms spanning a period of 6 months. The applicability of the

method has been demonstrated by tracing the propagation history of the North Rift, from the rift activation up to the calving of the iceberg A74.

For the North Rift, the shape of the delineated crack agrees well with the calving line location after the iceberg calving and with the rift location observed in optical imagery, thereby demonstrating the suitability of the approach. In general, phase artifacts in the interferograms may introduce noise in the delineated cracks, but the InSAR-based method is still less impeded by topography and structural heterogeneity of the ice shelf than SAR backscatter imaging. A limiting factor of the method is the decorrelation caused by snow drift, snow melt or fast-flowing ice, as it prevents any InSAR measurements.

In addition to the propagation history, the temporally variable phase contribution could be isolated and interpreted as rigid-body rotation about the expanding tip of the North Rift in response to the rifting activity. Without further information, it is not possible to determine whether the rotation origin is an ice flow speedup or the progressive separation of the future iceberg from the shelf.

Combined with the continuous 6-day coverage of the Antarctic margins by Sentinel-1, the InSAR-based crack delineation opens the possibility for operational monitoring of damage and rifting activity over most ice shelves, as well as the detection of ongoing breakoff processes and precursor signs of calving events. Thanks to the high sensitivity of InSAR to dynamic changes in the ice shelf strain pattern, the rifting activity can be captured well before it is visible in SAR backscatter images. The tip of the crack is also observed a few days in advance compared to optical images. With SAR interferometry, the quasi-full propagation of the North Rift could be mapped from image pairs acquired about a month before the iceberg broke off. This is a major advantage for predicting future calving events and to improve modelling of the response of ice shelves to damage.

Future work should focus on testing the method on other ice shelves, with various structures and geometries, on improving the post-processing for reducing the errors in the detected cracks and on developing synergies with other detection methods.

*Data availability*. Sentinel-1 satellite data are freely available on the ESA Open Access Hub (https://scihub.copernicus.eu). Offset-tracking products can be downloaded from the ENVEO cryoportal (https://cryoportal.enveo.at/). Interferograms used for delineation and a shapefile of detected cracks can be downloaded at following URL: https://cryoportal.enveo.at/data/Brunt-Cracks.

*Author contributions*. LL developed the algorithm, performed the processing, led the data analysis and drafted the manuscript. JW and TN contributed to the data interpretation and the discussion of the results, and revised the manuscript.

*Competing interests*. The authors declare that they have no conflict of interest.

*Acknowledgements*. This work was carried out in the scope of the ESA Polar+ Ice Shelves project (4000132186/20/I-EF). The authors thank the Polar+ Ice Shelves team members for their insights and helpful discussions. The authors also thank the reviewers and the editor, Reinhard Drews, for their useful comments that contributed to improve the paper.

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
