# Peer review of "Automatic Delineation of Cracks with Sentinel-1 Interferometry for Monitoring Ice Shelf Damage and Calving"

_The Cryosphere, 2021_

## Referee Comment (RC1)

Comments on "Automatic Delineation of Cracks with Sentinel-1 Interferometry for

Monitoring Ice Shelf Damages and Calving".

Overall a well-written paper, which is generally easy to follow, although there are some minor points that should be clarified.

P.5, L116: *"cracks at the scale of the pixel resolution".* I think "spatial resolution" is a better term, as this is a lower limit (pixel size is often smaller).

P5, L116 to L123: Please provide numbers for the resolution, wavelength, and revisit time of the sensor, as this helps in interpreting the results and may not be known by everyone.

P5, L131, *"we assume the phase noise is negligible"*. Although it is not relevant exactly for the discussion here, the temporal decorrelation should be briefly discussed somewhere in the introduction or theory section.

P6, L139: The sign of $D_{tides}^{ij}$ used in equation (3) is not defined – is it positive for an upwards motion? In that case, the sign in the equation seems wrong. Since it is a purely vertical motion, perhaps it would be better to define it as a scalar.

P8, L193 *"The wrapped interferogram is geocoded"*. Would it not be easier and more precise to calculate the gradients and edge detection in the original radar geometry, where the resolution in each dimension is known and can be accounted for, and no geocoding of a wrapped phase is required? Then just the final result could be geocoded. Please comment on this.

P8, L201: "*The discrete phase derivatives are computed by averaging the phase differences between adjacent pixels along the x- and y-directions over a square window*". The windows applied in eq 6 and 7 are not square, they are one-dimensional.

P10, L224: "*we neglect the phase gradient direction*". Of course, the gradient direction is not meaningful when the magnitude is low and should not be used in this case, but could it not be useful in a situation where the magnitudes on two sides are equal, but the directions differ? Please comment on this.

P11, L254 "*uncompensated tidal displacements*". These have no component in the along-track direction so they should not lead to phase jumps at the burst overlaps?

P13, L291 "*all areas above 50 m height*". Is this a general rule or does it only apply to this dataset?

P17, Section 6.4: Could the interferometric coherence perhaps show some of the fractures more clearly than just the backscatter image? Please comment on this, and maybe provide an example if this is so. Maybe also comment on whether the interferometric coherence could add some value to the processing, other than just the thresholding.

P19: Figure 12: The figure comes before it is referenced in the text. The same goes for Figure 13.

P20, L398 and 402: "*Differentiation*" implies finding a derivative. Please use the word "differencing" or "difference between"

P20, L411: "*This number of fringes corresponds to a displacement of about 35 cm in the direction of the line-of-sight*". Isn't it technically a change in LOS displacement, changing along the fringe belt? What is the direction of the change (negative or positive in LOS)?

Some minor typos:

P1, L.10: "These unprecedented ... enable" should be "The unprecedented … enables"

P2, L.33 "results into" should be "results in"

P2, L.43 "iceshelves is" should be "iceshelves are"

P2. L.56 "wide SAR images" should be "wide swath SAR images"

P5, L105 "November 2021" should be "November 2020"

P5, L120 "deramping or burst stitching" should be "deramping and burst stitching"

P7, L180: "hence" should be deleted

P7, L184 "account for" should be "accounting for"

P12, L274 "REMA DEM" should be "the REMA DEM"

P21, L417: "opposite", please use another word, like. Opposite suggest a 180° change of the LOS direction.

---

## Author Response (AR1)

**Automatic Delineation of Cracks with Sentinel-1 Interferometry for Monitoring Ice Shelf Damages and Calving**

We thank the reviewers for their comments and corrections. We tried to respond in a relevant and concise way to all the comments. In the following, the referee's comments are reported in italic bold font. The replies of the authors are provided below each comment in normal font. We first address the comments of Referee #1 and then those of Referee #2.

Please mind that, resulting from the review process, one figure has been added and the numbering of the last two figures has therefore been shifted by one (Fig. 12 -> Fig 13 and Fig. 13-> Fig. 14). Some equations have also been added.

**Reply to Referee #1**

***Overall a well-written paper, which is generally easy to follow, although there are some minor points that should be clarified.***

- ***P.5, L116: "cracks at the scale of the pixel resolution". I think "spatial resolution" is a better term, as this is a lower limit (pixel size is often smaller).***

We agree and changed "pixel resolution" to "spatial resolution" (L114).

- ***P5, L116 to L123: Please provide numbers for the resolution, wavelength, and revisit time of the sensor, as this helps in interpreting the results and may not be known by everyone.***

We have added a table with the main sensor characteristics specific to S-1 SLC acquisition in IW mode (see Table 1).

- ***P5, L131, "we assume the phase noise is negligible". Although it is not relevant exactly for the discussion here, the temporal decorrelation should be briefly discussed somewhere in the introduction or theory section.***

We briefly introduced temporal decorrelation and the main factors causing it in L151-154.

- ***P6, L139: The sign of $Dtides^{ij}$ used in equation (3) is not defined – is it positive for an upwards motion? In that case, the sign in the equation seems wrong. Since it is a purely vertical motion, perhaps it would be better to define it as a scalar.***

Indeed, in this case, there is an error in equation (3). We changed the vector $\vec{D}_{tides}^{ij}$ to a scalar parameter $D_{tides}^{ij}$ and specified that upward motion corresponds to positive values of the parameter. This corresponds to a negative change in slant range when projected on the LOS and therefore a negative phase component. We change the sign in equation (3) accordingly.

- ***P8, L193 "The wrapped interferogram is geocoded". Would it not be easier and more precise to calculate the gradients and edge detection in the original radar geometry, where the resolution in each dimension is known and can be accounted for, and no geocoding of a wrapped phase is required? Then just the final result could be geocoded. Please comment on this.***

The phase gradient and derived products could indeed be calculated in the original radar geometry. However, for the following reasons, we prefer the geocoded approach:

1) Pixels usually do not have the same dimensions in the slant range and azimuth directions and it is therefore necessary to apply a scaling factor for calculating the gradient (i.e. calculate the

phase variation per meter or per degree, not per pixel). In this case, one can easily consider different length scales for calculating the spatial gradient in both directions, which should be avoided. In the geocoded case, pixels are usually and easily squared, making it straightforward to work at the same scale in both directions.

2) Although we do not use the phase gradient direction in this context, it seems more natural to calculate it in geocoded geometry and provide an angle relative to a projection axis, rather than to calculate it in radar geometry and provide an angle relative to the satellite flight path. Generally speaking, it should allow to work with gradients calculated along different orbits.

- ***P8, L201: "The discrete phase derivatives are computed by averaging the phase differences between adjacent pixels along the x- and y-directions over a square window". The windows applied in eq 6 and 7 are not square, they are one-dimensional.***

There was indeed a mistake in the mathematical description of the phase gradient calculation. A second summing sign, accounting for the second dimension of the window, has been introduced in equations (6) and (7) for describing adequately the calculation that is performed in practice (L217-218).

- ***P10, L224: "we neglect the phase gradient direction". Of course, the gradient direction is not meaningful when the magnitude is low and should not be used in this case, but could it not be useful in a situation where the magnitudes on two sides are equal, but the directions differ? Please comment on this.***

One could imagine a situation where the fringe rate is similar on both sides of a crack, but fringes have a different orientation, in which case, yes, the phase gradient direction would be useful. However, exploiting the phase gradient direction would require an edge detector that would be able to deal with the wrapping of the angles and would still be as efficient as, for example, the Canny edge detector. Ideally, the information from both phase gradient magnitude and the phase gradient direction should be combined in order to exploit the comprehensive fringe information. Such investigations are ongoing but are still at an early stage. For the study case presented in this paper, the phase gradient magnitude seems to already provide an added value.

- ***P11, L254 "uncompensated tidal displacements". These have no component in the along-track direction so they should not lead to phase jumps at the burst overlaps?***

That is right and the sentence has been removed.

- ***P13, L291 "all areas above 50 m height". Is this a general rule or does it only apply to this dataset?***

It is specific to this dataset. The mask could also have been derived from the grounding line location, as elevation drops clearly in this region.

- ***P17, Section 6.4: Could the interferometric coherence perhaps show some of the fractures more clearly than just the backscatter image? Please comment on this, and maybe provide an example if this is so. Maybe also comment on whether the interferometric coherence could add some value to the processing, other than just the thresholding.***

The North Rift can be observed clearly in some coherence images. Other cracks like Chasm 1 or Halloween crack, not necessarily well-captured by the phase gradient, appear also clearly in coherence images. We provide a sample of coherence examples in Figure 1. Depending on the viewing geometry, the crevasses appear also highly contrasted.

[Figure]

Figure 1: Sentinel-1 coherence over Brunt Ice Shelf. Acquisition dates are annotated in the upper left corner of each image.

In practice, while interferometry may accommodate with coherence levels of about 0.4-0.5, which are quite often encountered, such coherence values reduce strongly the contrast between the ice sheet background and the crack in the coherence image. We observe it around the North Rift, e.g. for the acquisition of 12-18 November 2020. In the few examples provided here, we also observe that a crack may sometimes result in a positive contrast, and sometimes in negative one (see the Chasm 1 on 7-13 September compared to the other dates), depending on the changing conditions (snow, melt, wind, etc.) on the ice shelf. The variations of contrast level and contrast sign make it challenging to use coherence for crack detection and would require further characterization of the coherence behaviour.

In Section 6.4, we aim at comparing the information held by interferometric phase against imagery and show the benefit of the first over the latter. Since coherence is a measure of the interferometric phase quality, it basically holds the same information as the phase and displaying coherence images instead of backscatter images would not meet our purpose. However, following the suggestion of the

referee #2, we added a comparison with Landsat-8 optical images for validating the tip of the crack (see Figure 12).

- **P19: Figure 12: The figure comes before it is referenced in the text. The same goes for Figure 13.**

We change the position of these figures and we have no doubt that this kind of formatting issue will be handled properly by the editor, if the paper is accepted.

- **P20, L398 and 402: "Differentiation" implies finding a derivative. Please use the word "differencing" or "difference between"**

This has been corrected (L454 and L458).

- **P20, L411: "This number of fringes corresponds to a displacement of about 35 cm in the direction of the line-of-sight". Isn't it technically a change in LOS displacement, changing along the fringe belt? What is the direction of the change (negative or positive in LOS)?**

Yes, in the double difference interferogram, the amount of fringes corresponds to a change in LOS displacement between the two interferograms, relative to the point where you start counting the fringes (in this case, the origin of the crack) and this relative displacement change has the same magnitude for all the points along a given fringe. We agree that the sentence might be misinterpreted and we modified it to make it clearer.

Positive phase corresponds to a motion (or change in displacement) in the direction away from the satellite. In the double difference interferogram along track 50, on the region north of the rift, the phase increases from the expanding tip towards the origin of the crack (MIR). South of the rift, between the North Rift and the Halloween Crack, we observe a phase decrease in the same direction. This actually indicates distinct stress field variation on both sides of the North Rift.

As a response to your comment and those of the second referee, we clarified the sign of the phase in the double difference interferograms and lengthened the discussion in section 6.5.

**Some minor typos:**

- **P1, L.10: "These unprecedented ... enable" should be "The unprecedented … enables"**

Corrected (L10).

- **P2, L.33 "results into" should be "results in"**

Corrected (L33).

- **P2, L.43 "iceshelves is" should be "iceshelves are"**

Corrected (L43).

- **P2. L.56 "wide SAR images" should be "wide swath SAR images"**

Corrected (L61).

- **P5, L105 "November 2021" should be "November 2020"**

Corrected (L106).

- **P5, L120 "deramping or burst stitching" should be "deramping and burst stitching"**

Corrected (L121).

- ***P7, L180: "hence" should be deleted***

Corrected (193).

- ***P7, L184 "account for" should be "accounting for"***

Corrected (L197).

- ***P12, L274 "REMA DEM" should be "the REMA DEM"***

Corrected (L305).

- ***P21, L417: "opposite", please use another word, like. Opposite suggest a 180° change of the LOS direction.***

We change it for "… a line-of-sight rotated by about 60° with respect to track 50" (L482).

**Reply to Referee #2**

*Libert et al. map the growth of a rift on the Brunt Ice Shelf, making use of the high frequency of Sentinel-1 imagery to provide a time series of crack evolution. Their interferometric method using edge detection of phase gradient magnitudes has an advantage over interferometrically-derived strain fields in that it does not rely on phase unwrapping, however, it has a disadvantage over other rift detection techniques (for example backscatter contrast or edge detection in optical imagery) in that it requires multiple images, with good coherence between image pairs. The method accurately tracks the location of a rift on the Brunt Ice Shelf. It also approximates the timing of rift growth. The timing is not validated by other observations, so it is currently difficult to assess the accuracy in detection of the rift tip itself.*

*The delineation of the rift is dependent on multiple (presumably tuned) threshold parameters, various stages of filtering and line cleaning to reduce unwanted noise. This appears to be a careful balance between keeping real cracks and removing artefacts. It would be valuable if the authors explained how the parameters are determined.*
*The paper is well written and provides a useful dataset for this particular ice shelf, but whether it would be valuable to apply to automatic crack detection on other ice shelves, given the added complexity and data processing requirements associated with interferometry and the lack of evidence that there is any improvement in positional accuracy, or detection success, is not clear.*

*Specific comments:*

- *Line 23: I don't think you should link fracturing and damage development to climate warming here. The large independent rifts you observe on the Brunt Ice Shelf are not related to climate warming and I have some doubts that your method would work well in areas where this is the case, for example where these is dense damage / cross-cutting cracks or complex shear margins.*

We agree that this is not demonstrated in the paper and that the method might not be applicable for any type of damages, especially over fast-flowing ice shelves with complex disintegration pattern at the calving front. A typical non-working example would be Pine Island Glacier. The reference to climate warming has been removed, both in the abstract and the conclusion.

- *Line 43: sp '…the majority of ice shelves are routinely monitored…'*

Corrected (L43).

- *Line 54-55: There is published work that suggests that SAR backscatter imagery can be used to detect narrow cracks under the right viewing geometry (e.g. Thompson et al., 2020, 10.1016/j.coldregions.2020.103128)*

Publications like Thompson et al. (2020) or Marsh et al. (2021, 10.1016/j.coldregions.2021.103284) do indeed suggest the possibility of identifying crevasses and cracks up to only a few centimeters wide (much smaller than the sensor resolution) using very high resolution TSX Stripmap (resolution of 1.2 m x 3.3 m) and Spotlight (resolution of 1.2 m x 1.7 m) mode imagery. However, these data are typically not acquired operationally and only cover small areas (e.g. 10x10 km for TSX Spotlight, 30x50 km for TSX Stripmap). Both studies also underline that crack/crevasse visibility is strongly dependent on the viewing geometry (look direction, incidence angle) and the crack orientation. Another strong

dependence arise from the water fraction in the snowpack, that reduces signal penetration when increasing and masks deeply buried features, especially during the melt season. Though these two publications highlight the potential for identifying crevasses with SAR backscatter imagery, they do not propose a detection method and the crack identification is performed in these studies by visual inspection. Moreover, the focus of our paper is set on active rifts, not on the detection of crevasses that may or may not be active.

For improving the state of the art, we have added the two references mentioned above in the introduction and described the pros and cons of SAR backscatter imagery for damage detection (L46-54).

- *Line 107: It is not necessary to repeat what you are about to say in the next section*

The sentence has been removed.

- *Line 223: I expect that in the event of a crack opening in pure mode I extension parallel to an ice shelf front, while the velocity may be different on either side of the crack, the phase gradient may not show a significant difference. In this case, edge detection on your offset tracking output may work better. Have you compared this to the interferometry?*

We have tested the edge detection on the velocity magnitude derived from offset-tracking. Doing so, the delineation results are less noisy and the North Rift can be detected, but not necessarily the other cracks. For example, Halloween crack is not always nor completely detected for the dates that we have considered. Though we acknowledge that the timing of the INSAR-based detection is not fully validated, it seems that interferometry captures a more advanced location of the crack tip compared to offset-tracking. This is in agreement with the higher spatial resolution of SAR interferometry and its better sensitivity to changes.

- *Line 249 – 258: The method appears to suffer both from false positives and false negatives. The Halloween Crack was active during this period (continuing to widen by almost 0.5 m / day in the center).*

Most of the "false detections" (false positives), e.g. close to the grounding line, originate from crevasses. We attempt to clean them because we aim primarily at mapping fractures and they make the detection results noisy, but these detections still picture actual damage. Discriminating between one type of damage and another is obviously challenging.

Regarding the Halloween crack, it is not fully delineated, but part of it is detected for some dates. A possible reason for missed/partial detection of Halloween crack could be that the widening does not introduce as much change in the strain field as the propagation. From the backscatter images, we observe no advance of the Halloween crack over this time period. Let us also mention that the test performed with offset-tracking did not allow to detect the Halloween crack either.

In order to avoid confusion, we rephrased the sentence at L288. Instead of "inactive", we say that the Halloween crack was "not propagating" at this time period.

- *Line 288: Does this 9 x 9 refer to the value for 'w' in the previous section? It would be useful to restate here (i.e. w = 9).*

It refers indeed to the *w* parameter of the previous section. We restated it, as suggested (L319).

- ***Paragraph 288: As the method is described as automated it is important to explain how these values were determined. What is the sensitivity of the results to these values? Do these parameters need to be changed if the velocity is different, or if the coherence is worse, or under a different viewing geometry, or on a different ice shelf?***

Detection parameters were determined by testing different sets of values on a given pair of acquisitions, and fine-tuned for a balance between detections and false alarms. The upper threshold is critical for detection, as it is the main driver for selecting edges. The value of the lower threshold is less critical, as it mainly allows the connection between the lines corresponding to the strong edges. The value of the detection parameters may need to be adapted in other cases.

The value of the detection threshold mostly depend on:

- Temporal baseline
- Velocity and global orientation of the velocity with respect to the viewing geometry (LOS)

To clarify this, we have added a paragraph (L263-278), showing that the Canny edge detection performs the thresholding on a quantity that is proportional to the temporal baseline, the incidence angle and the local absolute variations of the strain rate. If the expected strain rates are known or can be estimated, an approximation of the thresholds can be obtained. We have also added a sentence L325-329 for stressing that our choice our parameters was an empirical one.

- ***Figure 10: There are only 7 points on this graph, but you say that 32 interferograms were generated (including 9 not used due to coherence issues). It would be useful to add the other 16 points, even if the detected crack length does not change between interferograms. The crack moves in discrete jumps and any periods of no movement are also of interest. This would also help readers to assess the uncertainty in the method.***

For a better summary of the overall results, we have added a panel to Figure 9 that shows the delineation results at the ice shelf scale for all coherent pairs of acquisitions. We have updated figure 10 with the corresponding points. For some pairs, negative propagation rates are measured due to the inaccuracy of the method. In this case, we do not provide the estimated rate values since they are not relevant. We have also updated the text of Section 6.3 accordingly.

- ***Line 363: While it may be difficult to delineate the rift in Sentinel-1 backscatter, it was visible in Landsat-8 at a similar location to the interferometry on 19th January and almost fully visible to the Stancomb-Brunt Chasm by the 6th February, suggesting there is not necessarily a significant information gain using this InSAR method, relative to optical imagery.***

We have analysed these two Landsat-8 images and decided to add them to the discussion in section 6.4. The rift is indeed well visible in those images and the comparison provides a real added value to the paper.

We compared the InSAR detection of 17-23 January with the North Rift location in both Landsat-8 images. This InSAR detection is coincident with the Landsat images of 19 January. The InSAR-based detection shows an agreement within 200 m over most sections of the rift. It is however longer than

what is observed in the Landsat-8 images. Comparing the InSAR detection of 17-23 January with the Landsat images of 6 February that shows that crack at a later stage of expansion, we observe that the tip of the rift agrees with the curvature of the rift. This demonstrates the increased sensitivity of InSAR compared to imagery, since it captures the tip location a few days ahead the optical image.

These results have been added (see Figure 12) and the discussion of Section 6.4 has been completed with the analysis of the Landsat-8 images (L405-422).

- *Line 366: This sentence is a bit odd, consider rephrasing.*

We rephrased the sentence, as suggested (see L424-425).

- *Fig 12 (a-c): As the fringes are fairly close, it is not immediately obvious from the figures that the color order of fringes is reversed on one side of the crack with respect to the other. This is an important point to highlight that the change in velocity is opposite on either side of the crack (particularly given that the fringe frequency is similar). This could be mentioned in the text to draw the readers' attention.*

Indeed, the color order is reversed on both sides of the cracks. North of the crack, the phase increases from the tip of the crack towards the McDonald Ice Rumples, while the phase behaviour is reversed on the other side of the crack. This indicates distinct responses on both sides of the crack.

Following your suggestions and in response to the other referee's comments, we clarified the phase sign, discussed the phase behaviour in the double difference interferograms in more details and lengthened section 6.5.

- *Line 410: You should probably refer to strain rate (as they are being measured) rather than stress.*

We modified stress for strain in the text (L473).

- *Line 443: What do you mean by 'could be misinterpreted as ice flow acceleration by offset-tracking'?*

We aim at distinguishing between a drift motion of the entire chunk of ice, as a whole body, and the speedup of the fluid ice flow. Both would be physically different, especially in terms of vertical gradient (no gradient in the first case, shearing in the second one) that we cannot capture with interferometry or offset-tracking, but they would result in a similar signal since we see only the speed of a single horizontal layer. We attempt to clarify it in L512.

- *Line 449: The derived rift growth pathway agreeing with the final calving pathway does not fully demonstrate the suitability of the approach for understanding the timing of rift growth. Would it be possible to validate the derived rift tip via other means (e.g. optical imagery)?*

The comparison with final calving pathway aims at validating the spatial extent of the crack, though we know that it cannot account for e.g. secondary branches. Without in situ observations, the timing of the rift growth remains challenging to validate, especially for the tip of the rift, as different delineation methods (based on remote sensing) may have different sensitivities.

In order to strengthen the validation, we exploit optical imagery as suggested (the Landsat-8 images mentioned previously). We compare the crack delineated manually from the Landsat-8 figure and measure the distance between this reference and the edges delineated with INSAR at the same date. The results are provided in Figure 12 and discussed in Section 6.4.

- ***Line 456: Again I am not sure what you are referring to here with respect to ice flow speed up vs ice drift.***

See the reply to the previous comment.

---

## Author Response (AR2)

**Automatic Delineation of Cracks with Sentinel-1 Interferometry for Monitoring Ice Shelf Damage and Calving**

We thank the Editor for its comments and corrections. In the following, the editor's comments are reported in italic bold font. The replies of the authors are provided below each comment in normal font. The line number refer to the corrected manuscript without track change.

**Reply to Editor**

***Thank you for submitting your revisions which was positively evaluated in the re-review. The manuscript is now almost ready for publication. Congratulations!***

***Unfortunately, I do not concur with the reviewer's judgement of the high presentation quality. Below you will find many comments regarding language, definitions and layout of the manuscript.***

***In particular, I believe we do not have the same concepts/terminology to describe the different ice dynamic components. Although I understand your desire to differentiate between different mechanisms (e.g. "deformation" and "flow" in equation (1)), I don't think this is correctly done here (cf comment to L457). Also I believe that InSAR cannot differentiate between those components without external knowledge. This must be worked out more clearly.***

We apologize for the inaccurate terminology used for describing ice dynamics. Given the confusion raised by our choice to describe the different phase components, we decided to restrict ourselves to the tide and ice flow contribution, and to remove the deformation/displacement part that may be less relevant for a general ice shelf description. We do believe, however, that such a distinction can be made between ice flow and displacement in some circumstances: when the ice detaches from the shelf, the ice does not "flow", it "moves" (c.f. explanation of the rigid body rotation in section 6.5).

Regarding the InSAR phase, we do not wish, in no instance, to let readers think that InSAR can a priori differentiate between the different phase components. We attempt to clarify this in the manuscript by using words such as "blend", "combination" or "cumulative effect".

***Another example is that "stress", "strain" and "strain rates" appear to be used interchangeably but those are surely not the same things.***

The words "stress", "strain" and "strain rates" have indeed been loosely used and confused with each other. We revised their use throughout the manuscript, which should now be adequate.

***None of my comments challenge the concepts or findings of the paper, but I still urge them to take them seriously so that the paper can be made more accessible to a wider readership.***

- **L 56 remove „for the first time". This also doesn't fit with the reference from 2016.**

The paper by Torres et al. (2016) discusses the data availability, coverage and revisit of Sentinel-1. It presents the global observation scenario of Sentinel-1 and it supports the fact that this mission is a gain for continuous monitoring compared to previous missions. We believe that "for the first time" is adequate as such a long-term systematic coverage of Antarctica was never achieved with SAR sensors before Sentinel-1.

- **L 68 "about" -> "around" (?)**

Both "about" and "around" are correct in this context.

- **L 74 InSAR does not detect "stress field changes", only if those lead to deformation. Consider rephrasing.**

We rephrased the sentence (L74-75).

- **L 95 I don't think the abbreviation SWIT is needed.**

The abbreviation has been removed.

- **L 110 Again I don't think InSAR captures "stress". It captures strain rates related to stress gradients.**

You are right. InSAR does not directly capture the "stress", it captures the strain (deformation) resulting from the stress – if the deformation is not spatially constant. The difference between two consecutive interferograms therefore captures the variation of strain pattern. We rephrased the sentence (L111).

- **Figure 1: Add text "Fig 1c,d" to the white dashed box in Fig 1a.**

Figure 1 has been modified.

- **L 140 I don't see icebergs being treated here. Remove this example.**

Modified (L143).

- **L 140 Somehow, I feel this is described more complicated than needed. How can you differentiate in an interferogram between $d^{ij}_{LOS}$ and $v^{ij}_{LOS}\Delta^{ij}$ ? I don't think you can.**

No, indeed, you cannot differentiate both without further information. As stated in the response to your previous comment, we simplified the description and removed the deformation/displacement component for the general description of an ice shelf (see Equation 1).

- **Fig 2a does not show "tidal bending" (nothing is bend here). Replace with "vertical tidal displacement". Bending only occurs in the grounding zone.**

We changed "tidal bending" for "tides" (L157).

- **L 166: add "..SAR interferometry captures [the cumulative effect] of spatially…". Rephrase the stress field component.**

We rephrased the sentence (L166-167).

- **L 168 In order for tides to be visible (rather than a constant phase offset) you require a spatially variable tilting of the ice shelf, correct? I don't understand how you infer that "…the change of vertical position for a given pair of acquisitions shows smooth spatial variations of only a few centimeters". The interferometric phase lumps all effects (ice-shelf tilting through tides, horizontal ice flow motion,..) together so how do you infer without external evidence that one of the components is small? I agree that horizontal ice-flow is more important than a residual tidal signal in the vertical but the chain of arguments for this is unclear.**

Yes, the tides are captured by the interferograms if the surface is tilted. That is what we intend to say when talking about "spatial variations".

It is of course impossible to infer the amplitude of vertical displacements caused by tides directly from the interferogram. This information comes from the same models that we use for tidal correction of

offset-tracking maps, i.e. CATS2008 for tides and ERA-5 for atmospheric pressure. We realized that it may be unclear from the text and we rephrased it (L172-173).

- **L 177 "…crack detection is performed on interferograms containing the information of all three phase components". This sentence and much of the information in the paragraphs above suggests that you can in theory differentiate between the first three phase components in eq. (1) in an interferogram. I don't think that this can easily be done (e.g., you would need a model for ice-shelf tilting as a response to spatially variable tides).**

We actually intend the exact opposite. As we cannot discriminate the different phase contributions and we do not know for sure what are the processes involve in the rifting activity, we cannot establish the origin of the interferogram segmentation. Therefore we assume that the discontinuity may arise from each one of the components. We rephrased the paragraph in order to clarify this (L179-183).

- **L 200 Which DEM is used for the topographic phase correction? What is the effect if the DEM does not capture small-scale variability such as cracks?**

The TanDEM-X polar DEM with 90m spatial resolution is used for subtracting the topographic phase. Obviously, surface features such as crevasses and cracks advancing with the ice flow can hardly be subtracted by a DEM.

In practice, the SAR signal sees only some part of the crack depth, if not in the shadowed area, because the crevasse walls are steep and very high incidence would be needed to see the bottom of the crevasse. The subtraction of a DEM that does not capture the crevasse surface roughness will still leave a residual topographic phase and introduce local phase discontinuity. Contrary to rifts and cracks, we do not usually observe phase ramps with different orientations and fringe rate on both sides of the crevasses. We added a short discussion on this effect (L168-171).

- **L 204 "The phase signal in the interferogram is a sum of the ice motion component, the tidal component and the random phase noise…" . Eq (1) lists four terms, here only three are mentioned. Does this mean one component (supposedly \phi^{ij}_{flow} ) has been removed? I don't think this is the case. The more I advance through the paper, the less convinced I am that the distinction of \phi^{ij}_{flow} and \phi^{ij}_{defo} is needed.**

See our previous reply on the simplified INSAR signal model.

- **L 240 All tuning parameters should be explicitly mentioned so that the study is reproducible.**

All parameter values used for this experiment are clearly stated in Section 5. Since they may change from one test site to another, we prefer not to specify them in the "method" section, as it could make the reader think that those parameters are directly transferable to other cases.

- **L 247 What is the reason that the 50 m contour line of the TanDEM X DEM is used rather than any published grounding line (e.g., the one displayed Fig 1a)? The clipping will be unnecessarily inaccurate.**

Grounding line locations are usually provided as vector lines, very often discontinuous, making it not straightforward to convert into a mask. They also do not necessarily fit the location of the grounding line during period of interest. Given that the region around the Brunt Ice Shelf grounding line is highly crevassed and that a comparison with the NASA MeaSUREs' grounding line location showed no major change of the mask coverage, we decided to stick to the DEM-based mask. This decision was also supported by the fact that the focus of the study is set on the North Rift far from the grounding line location.

- **L 255 Provide numbers for "small" to keep everything reproducable.**

As for the comment on L240, the cutoff length of the dangles is given in Section 5 (L353).

- **L 319 Here the parameters are provided that I asked for previously. I think this can be presented more succinctly by including the "processing" steps directly in the methods.**

As answered to the comment on L240, we prefer to provide the parameter values altogether in a section describing the experiment, as we do not wish the reader to believe that these parameters are necessarily valid for all study cases. We also believe that, if the reader wishes to reproduce the results, it is easier for him/her to have all the parameters listed at the same place rather than looking for them throughout the text.

- **L 363 "As already stated above, …" no need to start the sentence like this. Either its worth repeating or it is not.**

We rephrased it as: "As observed also in Fig. 8…" (L393).

- **L 428 remove "))" and "see".**

This has been modified (L460).

- **Figure 10 What does the AA stand for in the title?**

"AA" stands for Antarctica. This has been changed to the full name.

- **L457 "..changes in flow velocity or creep deformation..". I believe we have different ice dynamic concepts. For me ice-flow has two components one due to internal ice-deformation and one to basal sliding. Both leads to a displacement at the surface. If "creep deformation changes" so does the "ice flow". I believe what you are trying to distinguish here is "plug flow" (which is the most relevant flow regime for ice shelves with essentially 100% basal sliding and 0% deformation) and other components. Anyway, the way this is written here and also elsewhere repeatedly confuses me.**

As already stated before, we reviewed the ice dynamics terminology throughout the text. For this specific comment, we removed "creep deformation" and left only "ice flow" (L205).

> **Later in that sentence only "ice flow" cancels during the doubled differencing. What happens to the "creep deformation" ? Further on in this sentence only the "natural variations…and the deformation" are left in the double differenced interferograms. What do you mean by "natural variations" ? Tides go "naturally" up and down, but this is not what is left in a double differenced interferogram. What is left is the differential tidal signal.**
> **How about something like: "The differential phase of double differenced interferograms is composed out of the differential vertical displacement by tides and time-variable components in the horizontal velocities between the two interferograms.**

The sentence has been rephrased (L204-207).

- **Section heading 6.5: Rephrase, InSAR does not see stress field variations. It sees related strain rate patterns.**

The heading has been modified.

- **This entire section starts be reiterating basic concepts of differential interferometry. This should be moved to the methods.**

We re-arranged Section 3 on SAR interferometry over ice shelves and move the first part of Section 6.5 to a subsection of the Section 3.

We choose to move it to Section 3, which is more about "rationale", instead of Section 5 because this paragraph is part of the result interpretation, not of the detection method.

- **L430 "bulk contribution" seems like an unfortunate term. Define it as "the time-invariant contribution" or something like that.**

As suggested, we replaced it as the "time-invariant core contributions" (L181).

- **L 470 Differential interferometry does not "isolate" the signal caused by rifting activity. As stated in the paragraphs before this differential signal still contains other processes (e.g. differential tides). Maybe "amplify"?**

We changed it to "emphasize" (L473d).

- **L 473 The first interferogram sees the "strain" the second "the stress field" and the third the "fringe pattern". This is another example where I urge you to use consistent terminology. Please unify the discrepancy of "stress", "strain", and "strain rates" throughout the manuscript. Those terms cannot be used interchangeably. I don't think InSAR can pick up any stress patterns.**

As already replied to your previous comments, we review the use of those words throughout the manuscript.

- **L 480 "..the observed phase ramp could either correspond to a vertical deformation or to a horizontal displacement..". I agree, but what about the deformational component that you introduced in eq (1,12)?**

See our previous reply on the modification of the INSAR phase signal model.

- **L486 Use "the differential phase" when you refer to double differenced interferograms.**

We are not certain to spot the confusion in this case. For the sake of clarity, we have specified "double difference interferograms" and added "differential phase" throughout the paragraph. In order to avoid confusion, we decided to switch from "differential interferograms" to "flattened interferograms" throughout the text.

- **L502 "dominates.." -> "..that is larger than the differential tidal displacement in the vertical."**

Corrected (L514-515).

- **L526 "iceberg drift"? Icebergs are fully disconnected from the ice sheet. I don't think this is the case here. Rephrase.**

We rephrased it as "…the displacement of the future iceberg as it separates progressively from the ice shelf after the opening of the North Rift over the full ice thickness and its consequent widening" (L525-526).

- **Acknowledgements: Consider thanking the reviewers.**

The acknowledgment has been added.